# Coverage ≠ Exposure: Auditable Control of Same-Support Tail Failures under Multimodal Missingness

Ziteng Hong [1]  Bingzhi Chen* [1]  Yishu Liu [2]  Sudong Cai [1]  Guangming Lu [2]

## Abstract

Multimodal systems encounter partial observability from sensor dropout and degradation, yet adverse tails can remain unreliable even when average performance is preserved. Under a locked same-support contract, matched-average runs can exhibit same-support tail failure, diverging on worst-case and shift-reweighted metrics over the same observable support. The failure arises because conditional interaction separates environment coverage from parameter exposure. Gated routing sends information through different groups, so high-leverage groups may receive few updates despite complete coverage, and tail aggregation amplifies their errors. We audit this mismatch with TailPressure, an exposure-normalized statistic of tail-leveraged interaction from gating logs. We propose Heterogeneity-aware Closed-loop Exposure Stabilizer (H-CES), a lightweight controller that stabilizes per-group pressure via deterministic increment-branch gating and group-wise decoupled weight decay, without changing loss or inference. Across multimodal settings and backbones, H-CES improves same-support tail reliability while preserving clean performance.

## 1. Introduction

Multimodal systems are increasingly evaluated and deployed under partial observability. Sensors may drop out, saturate, clip, or be masked by policy filters, while many pipelines still record a compact health signal describing which modalities are present and how reliable each observed channel is. In these settings, the clean average is not enough: adverse health states can dominate downstream reliability

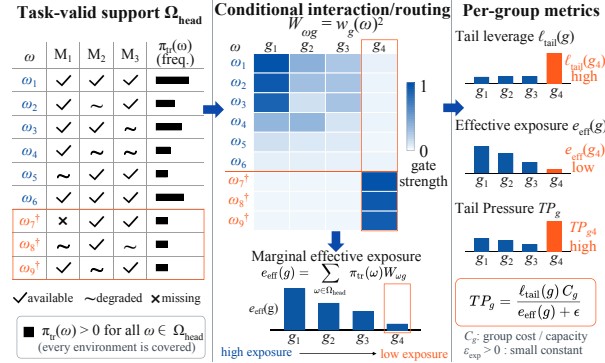

(a) Coverage Does Not Guarantee Exposure

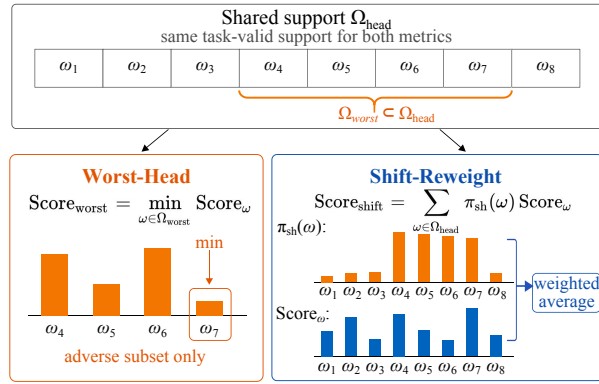

(b) Tail Metrics on the Same Support

*Figure 1.* Locked same-support contract and the coverage–exposure mismatch. (a) distinguishes environment coverage from parameter exposure under conditional interaction, and shows how a contract-fixed tail leverage coefficient can amplify underexposed groups. (b) sketches the two headline tail aggregates evaluated on the same task-valid support.

even when headline performance appears stable (Hendrycks & Dietterich, 2019; Taori et al., 2020; Koh et al., 2021). Figure 1(a) separates environment coverage from parameter exposure, and Figure 1(b) shows the fixed-support tail functionals used throughout the paper.

Existing robustness tools mainly intervene at the data level or at the aggregate objective. Missingness augmentation enumerates dropout and degradation patterns during training

[1]Beijing Institute of Technology, Zhuhai, China [2]Harbin Institute of Technology, Shenzhen, China. Correspondence to: Bingzhi Chen* <chenbingzhi@bit.edu.cn>.

*Proceedings of the 43rd International Conference on Machine Learning*, Seoul, South Korea. PMLR 306, 2026. Copyright 2026 by the author(s).

*Table 1.* Same-support pairing statistics on $\Omega_{\text{head}}$ (MMBench-EN dev v1.1, LLaVA-NeXT/Vicuna-7B; 36-config scan, 1 seed). Eligible pairs match Full and Train-Avg within $\delta_{\text{full}} = \delta_{\text{avg}} = 0.2$. Worst-Head uses contract $\Omega_{\text{worst}}$, Shift-Reweight uses contract $\pi_{\text{sh}}$, and Random-Worst is a same-size same-support control.

| Tail functional | median $|\Delta|$ | q90 $|\Delta|$ | frac($|\Delta| > \text{CI}$) |
|---|---|---|---|
| Worst-Head | 4.3 | 9.0 | 0.72 |
| Shift-Reweight | 3.3 | 6.6 | 0.67 |
| Random-Worst | 1.0 | 2.1 | 0.19 |

(Neverova et al., 2016), while tail objectives reweight examples or environments (Rockafellar & Uryasev, 2000; Sagawa et al., 2020). Recent multimodal benchmarks broaden the evaluated capabilities of large models, but comparisons often remain tied to compact headline aggregates (Fu et al., 2025; Yu et al., 2024; Liu et al., 2024b). This leaves a basic question unresolved: **If two runs have the same observable support and the same average performance, must their adverse-tail behavior also be close?**

We study this question with a locked same-support contract. Each example receives an observable environment label $\omega$ from modality availability and measurable quality. The contract fixes a task-valid support $\Omega_{\text{head}}$ before method comparison, computes Full, Train-Avg, Worst-Head, and Shift-Reweight on the support, and keeps stress-only environments outside headline claims. This separation is important for MNAR-sensitive regimes, where full-data quantities can be non-identifiable from observed data without additional assumptions (Rubin, 1976; Little & Rubin, 2019).

Under this contract, matched averages do not determine tail reliability. We pair scan runs whose Full and Train-Avg scores differ by at most fixed tolerances and then measure their Worst-Head and Shift-Reweight gaps on the same $\Omega_{\text{head}}$. Table 1 shows that contract-defined tail gaps remain large, whereas a same-support Random-Worst control collapses. Figure 2(a) visualizes the matched-pair scan, and Figure 2(b) shows that the divergence concentrates on recurring adverse environments within the same $\Omega_{\text{head}}$ rather than spreading uniformly across the support.

The mechanism we examine is conditional interaction. A training mixture can cover environments, while a gated multimodal model routes different environments through different parameter groups. A high-leverage group may therefore appear in the contract support yet receive few effective updates. We formalize this coverage-exposure distinction, derive the $w_g(\omega)^2$ scaling that enters group information, and define TailPressure as an auditable, leverage-weighted, exposure-normalized state computed from gate logs and checkpoint norms. Figure 2(c) previews the score-level bridge, and Figure 2(d) previews the alignment counterfactual. Appendix B.3 records the claim map that separates the contract, diagnostic, bridge, and controller evidence.

H-CES turns this diagnostic into a lightweight training-time controller. It uses a deterministic gate that scales only increment branches, reads exposure and strength from the training trace, and regulates per-group TailPressure through decoupled weight decay (Loshchilov & Hutter, 2019). The gate is replayable from observable health signals, the task loss is unchanged, and the inference interface remains the base multimodal model plus learned increments. We instantiate groups with adapters and low-rank updates (Houlsby et al., 2019; Hu et al., 2022), but the controller only requires groupable conditional interaction and an auditable gate for each group. Our main contributions are summarized below:

- We identify same-support tail failure, showing that matched-average multimodal runs can diverge on adverse tails even under the same observable support.
- We introduce TailPressure and H-CES, a replayable closed loop that links logged exposure, tail leverage, and group strength through per-group decoupled weight decay.
- We validate the mechanism through same-support comparisons, alignment-breaking counterfactuals, multi-backbone vision-language routes, sentiment routes, and non-LoRA interfaces, with gains concentrated on adverse-tail metrics while Full and Train-Avg remain stable.

## 2. Related Work

### 2.1. Multimodal Learning with Imperfect Data

Multimodal models fuse heterogeneous signals through interaction modules such as attention-based fusion and multimodal Transformers (Vaswani et al., 2017; Tsai et al., 2019). Robustness to sensor dropout and quality variation is commonly approached by injecting missingness or corruption during training to reduce reliance on any single modality (Neverova et al., 2016). Robustness benchmarking for adaptation of vision–language models under multimodal corruptions has also been studied at scale, highlighting that average gains can be uneven across corruption regimes (Chen et al., 2023). Missing data theory formalizes when full-data quantities are not identifiable from observed data, and MNAR mechanisms can induce different full-data risks while matching the observed distribution without additional assumptions (Rubin, 1976; Little & Rubin, 2019). This paper uses missingness theory only as a boundary statement and restricts mechanism claims to a task-valid support that is fully observable in the contract.

### 2.2. Tail Risk and Environment-Based Evaluation

Tail criteria such as worst-group objectives and conditional value at risk (CVaR) formalize protection of adverse parts of a distribution (Rockafellar & Uryasev, 2000; Sagawa et al.,

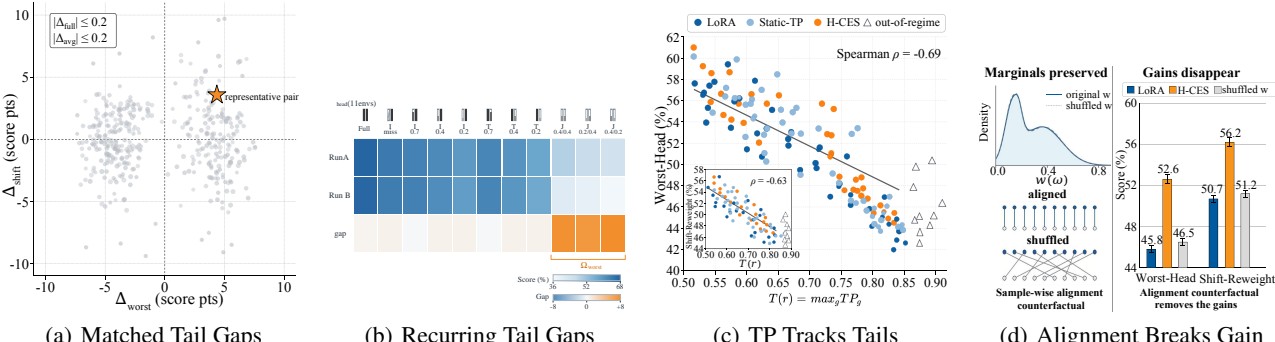

*Figure 2.* Same-support tail failures and auditable signatures on $\Omega_{\text{head}}$. (a) Matched pairs retain nearly identical Full and Train-Avg scores but exhibit tail gaps. (b) The gaps concentrate on recurring adverse environments. (c) TailPressure tracks headline tails within the audited regime. (d) Breaking gate alignment preserves marginals but removes the gain.

2020). Distribution shift benchmarks show that headline averages can conceal failures tied to environments (Taori et al., 2020; Koh et al., 2021). Recent multimodal evaluation suites often report many capabilities but still rely on compact headline numbers for comparison (Fu et al., 2025; Yu et al., 2024). This paper differs by fixing the evaluated environment support and studying a training-time instability that appears even when the headline support is held fixed.

### 2.3. Conditional Interaction, Grouped Interfaces, and Training-Time Diagnostics

Conditional computation routes inputs through different parameter subsets, including sparsely routed Mixture-of-Experts and its scaling variants (Shazeer et al., 2017; Lepikhin et al., 2021; Fedus et al., 2022). Parameter-efficient adaptation provides grouped trainable increments through adapters and low-rank updates (Houlsby et al., 2019; Hu et al., 2022). Recent variants adjust budgets or parameterizations while preserving a grouped interface (Dettmers et al., 2023; Zhang et al., 2023; Liu et al., 2024a). Diagnostic work has also used auditable statistics computed from gradients or activations to characterize training dynamics and guide schedules under fixed computation (Gray et al., 2024). This paper uses grouped interfaces as an auditable interaction surface and uses an exposure statistic aligned with a deterministic gate to diagnose same-support tail failures.

## 3. Locked Same-Support Contract

The locked same-support contract fixes a write-once mapping from an observable environment signal to a task-valid evaluation support and to tail aggregates computed on that same support. All headline comparisons use the same $\Omega_{\text{head}}$, and all mechanism claims are restricted to this set. An environment is an observable health signal. Let $M = |\mathcal{M}|$ be the number of modalities.

$$\omega = (b, q), \quad b \in \{0, 1\}^M, \quad q \in [0, 1]^M, \\ b_m = 0 \Rightarrow q_m = 0. \tag{1}$$

where $b$ indicates modality availability and $q$ summarizes measurable quality. Let $\Omega_{\text{test}}$ denote the finite set of environments enumerated by the contract.

The headline same-support set used in all headline metrics is the task-valid subset of $\Omega_{\text{test}}$ that satisfies a write-once validity predicate and a quality floor:

$$\Omega_{\text{head}} = \{\omega \in \Omega_{\text{test}} : \text{Valid}(\omega) = 1, \\ \min_{m: b_m(\omega)=1} q_m(\omega) \geq q_{\min}\}. \tag{2}$$

Throughout, the contract is instantiated so that $\text{Valid}(\omega) = 1$ implies at least one required modality is present, making the minimum well-defined. Stress-only environments are $\Omega_{\text{stress}} = \Omega_{\text{test}} \setminus \Omega_{\text{head}}$. Stress-only regimes can include task-invalid missingness or hard-missing settings that can be missing not at random (MNAR)-sensitive (Rubin, 1976; Little & Rubin, 2019). The stress-only metrics are reported separately and are excluded from headline claims.

We report a bounded sample-level score $s(\hat{y}, y) \in [0, 1]$ and the per-environment score:

$$\text{Score}_\omega(\theta) = 100 \cdot \mathbb{E}_{(x,y) \sim \mathcal{D}_\omega} \big[ s(f_\theta(x, \omega), y) \big] \in [0, 100]. \tag{3}$$

The contract fixes a full environment $\omega_{\text{full}} = (\mathbf{1}, \mathbf{1})$, a training mixture $\pi_{\text{tr}}$ supported on $\Omega_{\text{head}}$, a worst-case subset $\Omega_{\text{worst}} \subseteq \Omega_{\text{head}}$, and a shift reweighting $\pi_{\text{sh}}$ supported on the same $\Omega_{\text{head}}$. We report four headline aggregates computed on $\Omega_{\text{head}}$:

$$\text{Score}_{\text{full}}(\theta) = \text{Score}_{\omega_{\text{full}}}(\theta), \tag{4}$$

$$\text{Score}_{\text{tr-avg}}(\theta) = \mathbb{E}_{\omega \sim \pi_{\text{tr}}} \big[ \text{Score}_\omega(\theta) \big], \tag{5}$$

$$\text{Score}_{\text{worst}}(\theta) = \min_{\omega \in \Omega_{\text{worst}}} \text{Score}_\omega(\theta), \tag{6}$$

$$\text{Score}_{\text{shift}}(\theta) = \mathbb{E}_{\omega \sim \pi_{\text{sh}}} \big[ \text{Score}_\omega(\theta) \big]. \tag{7}$$

Worst-Head emphasizes the single worst environment in the write-once subset $\Omega_{\text{worst}}$. Shift-Reweight emphasizes a structured reweighting over the same support that reflects a deployment preference over adverse but task-valid regimes.

The contract objects are label-free and deterministic. The environment tables, tie-break rules, and the shift-family grids are pre-registered and shared across methods. The complete single source of truth (SSOT) tables appear in Appendix B.4 as the contract reference.

## 4. Phenomenon of Same-Support Tail Failures

This section documents a tail instability that persists under fixed support and matched averages.

We pool a run set $\mathcal{R}$ from a matched-budget scan. A pair $(a, b)$ is eligible when the two runs match Full and Train-Avg within fixed tolerances:

$$\left| \text{Score}_{\text{full}}^{(a)} - \text{Score}_{\text{full}}^{(b)} \right| \le \delta_{\text{full}}, \tag{8}$$

$$\left| \text{Score}_{\text{tr-avg}}^{(a)} - \text{Score}_{\text{tr-avg}}^{(b)} \right| \le \delta_{\text{avg}}. \tag{9}$$

The eligibility rule isolates a tail-specific instability by controlling for two headline averages computed on the same $\Omega_{\text{head}}$. For each eligible pair we measure the tail gaps

$$\Delta_{\text{worst}} = \text{Score}_{\text{worst}}^{(a)} - \text{Score}_{\text{worst}}^{(b)}, \tag{10}$$

$$\Delta_{\text{shift}} = \text{Score}_{\text{shift}}^{(a)} - \text{Score}_{\text{shift}}^{(b)}. \tag{11}$$

Table 1 reports quantiles of these gaps over all eligible pairs and the fraction of pairs whose absolute gaps exceed environment-stratified bootstrap uncertainty.

Worst-case operators are sensitive to variance, so we include a same-support control that preserves the aggregation form and the support while breaking the contract-defined structure. Random-Worst draws a random subset of environments with the same size as $\Omega_{\text{worst}}$ and applies min aggregation on $\Omega_{\text{head}}$. The gaps collapse under this control in Table 1, while contract-defined Worst-Head and Shift-Reweight remain large. This control removes an explanation that attributes the divergence to the min operator alone.

The tail divergence is not diffuse across $\Omega_{\text{head}}$. Per-environment matrices show that gaps concentrate on a small and recurring subset of adverse environments, as illustrated in Figure 2(b). This stability motivates a mechanism question. Two runs can match average performance on the same support yet still diverge on a recurring subset of adverse environments inside that support.

Pairing is computed over the entire run pool $\mathcal{R}$ and does not depend on the selection rule used for the headline tables. The headline tables in Section 7 compare methods using a single selected checkpoint per method under identical budgets and a fixed Full-drop constraint.

## 5. When Coverage Does Not Imply Exposure

This section formalizes a mechanism that is compatible with the phenomenon in Section 4. The mechanism distinguishes environment coverage from parameter exposure under conditional interaction. The formal analysis uses a surrogate risk and yields an auditable diagnostic state. The paper does not claim that surrogate-risk bounds imply bounds on the reported task score. The diagnosis is interpreted through an empirical bridge within an audited regime.

### 5.1. Coverage and Exposure as Distinct Objects

Coverage is specified by the training mixture $\pi_{\text{tr}}$ supported on the fixed headline set $\Omega_{\text{head}}$. Exposure quantifies how much effective information the optimizer receives about a parameter group under the realized training stream. Even with fixed $\pi_{\text{tr}}$, step-level exposure is stochastic from batch sampling, and group strength evolves during optimization.

The analysis uses a differentiable surrogate loss $\ell(\cdot, y)$ and the per-environment surrogate risk

$$\mathcal{R}_\omega(\theta) = \mathbb{E}_{(x,y) \sim \mathcal{D}_\omega} \left[ \ell(f_\theta(x, \omega), y) \right]. \tag{12}$$

All theoretical quantities in this section refer to $\mathcal{R}$. Section 7 evaluates a bridge between the diagnostic and the score tails defined by the contract.

### 5.2. Groupable Conditional Interaction and Deterministic Gating

We consider a frozen backbone with trainable increments partitioned into groups:

$$f_\theta(\cdot, \omega) = f_{\text{base}}(\cdot, \omega) + \sum_{g \in \mathcal{G}} w_g(\omega) \, \Delta f_g(\cdot; \theta_g), \tag{13}$$

where each group $g \in \mathcal{G}$ corresponds to an optimizer parameter group. The gate $w_g(\omega) \in [0, 1]$ depends only on observables and scales only the increment branch. This interface is used for auditability. The mechanism requires groupable conditional interaction, and a deterministic gate provides one sufficient instantiation.

**Definition 1** (Dependency set $\text{Req}(g)$). For a group $g \in \mathcal{G}$, $\text{Req}(g) \subseteq \mathcal{M}$ is the set of modalities whose tokens are direct inputs to the module(s) associated with $g$ under the fixed injection map.

We use a conservative monotone gate that requires all dependent modalities:

$$w_g(\omega) = \left( \prod_{m \in \text{Req}(g)} b_m \right) \cdot \min_{m \in \text{Req}(g)} q_m. \tag{14}$$

Because the gate is a deterministic function of $\omega$ and the dependency map, the gate is replayable from logs.

Exposure is measured at the optimizer-step time axis. Let $\mathcal{B}_t$ denote the global batch at optimizer step $t$ after gradient accumulation and across distributed data parallel (DDP) workers, with total size $B_{\text{glob}}$. Define the step-aligned mean

$$\overline{w^2}_t(g) = \frac{1}{B_{\text{glob}}} \sum_{i \in \mathcal{B}_t} w_g(\omega_i)^2, \tag{15}$$

and its exponential moving average (EMA)

$$e_{\text{eff},t}(g) = (1-\rho)e_{\text{eff},t-1}(g) + \rho \, \overline{w^2}_t(g), \tag{16}$$

with $\rho \in (0, 1)$.

### 5.3. Why $w^2$ Governs Information and Tails Amplify It

**Lemma 2** ($w^2$ scaling of gradients and curvature). *Consider a gated prediction*

$$z(\theta_g) = z_{\text{base}}(x, \omega) + w_g(\omega) \, \Phi_g(x, \omega)\theta_g, \tag{17}$$

*and a twice differentiable loss $\ell(z, y)$. Then*

$$\nabla_{\theta_g}\ell = w_g(\omega) \, \Phi_g^\top \nabla_z \ell, \tag{18}$$
$$\nabla_{\theta_g}^2 \ell = w_g(\omega)^2 \, \Phi_g^\top (\nabla_z^2 \ell)\Phi_g. \tag{19}$$

*Step-aligned effective information for group $g$ therefore scales with $\mathbb{E}[w_g(\omega)^2]$ under the realized training stream.*

We use a dimensionless strength proxy based on increment-to-base norms. For an injected layer $k$ in group $g$, let $W_k$ be the frozen base weight and $\Delta W_k$ the learned increment. Define

$$c_k = \frac{\|\Delta W_k\|_F^2}{(\|W_k\|_F + \delta_W)^2}, \tag{20}$$

$$\widehat{\mathcal{C}}_g = \frac{1}{|g|} \sum_{k \in g} c_k. \tag{21}$$

This proxy is auditable from model checkpoints and does not require access to optimizer state.

Tail aggregation amplifies groups through the gate. Because $\Omega_{\text{head}}$, $\Omega_{\text{worst}}$, and $\pi_{\text{sh}}$ are fixed by contract, we precompute a per-group tail leverage coefficient:

$$\ell_{\text{tail}}(g) = \max\left\{ \max_{\omega \in \Omega_{\text{worst}}} w_g(\omega)^2, \ \mathbb{E}_{\omega \sim \pi_{\text{sh}}}\left[w_g(\omega)^2\right] \right\}. \tag{22}$$

This coefficient quantifies how the headline tail functionals can amplify gated group perturbations without changing the evaluation support.

### 5.4. Incremental Surrogate-Risk Sensitivity and TailPressure

The following conditions are treated as auditable applicability conditions. Each condition has a logged proxy and

a fixed threshold rule that defines an audited regime used for mechanism evidence. The proxies and thresholds are reported in Appendix C.5.2.

**Applicability Condition 3** (Energy bridge, audited applicability). There exists $L_g$ such that for any sample $(x, \omega)$,

$$\|\Delta z_g(x, \omega; \theta_g)\|_2^2 \leq w_g(\omega)^2 L_g^2 \widehat{\mathcal{C}}_g, \tag{23}$$

where $\Delta z_g$ is the prediction perturbation induced by group $g$ at the loss input.

**Applicability Condition 4** (First-order residual control, audited applicability). In the audited regime,

$$\left| \mathbb{E}\left[ \langle \nabla_z \ell(z_{\text{base}}, y), \Delta z_g \rangle \right] \right| \leq \varepsilon_g^{(1)} \, \mathbb{E}\left[\|\Delta z_g\|_2^2\right], \tag{24}$$

for a small, log-estimable $\varepsilon_g^{(1)} \geq 0$.

**Proposition 5** (Incremental surrogate-risk sensitivity with tail leverage). *Assume $\ell(\cdot, y)$ is $\beta$-smooth on the audited domain and Conditions 3–4 hold. Then for any distribution $\pi$ supported on $\Omega_{\text{head}}$,*

$$\mathcal{R}_\pi(\text{with } g) - \mathcal{R}_\pi(\text{with } g \text{ off}) \leq K_g \, \mathbb{E}_{\omega \sim \pi}[w_g(\omega)^2] \widehat{\mathcal{C}}_g, \tag{25}$$

*where $K_g = (\varepsilon_g^{(1)} + \beta/2)L_g^2$. Under the contract:*

$$\Delta\mathcal{R}_{\text{tr-avg}}(g) \leq K_g \, \mathbb{E}_{\omega \sim \pi_{\text{tr}}}[w_g(\omega)^2] \widehat{\mathcal{C}}_g, \tag{26}$$
$$\Delta\mathcal{R}_{\text{shift}}(g) \leq K_g \, \mathbb{E}_{\omega \sim \pi_{\text{sh}}}[w_g(\omega)^2] \widehat{\mathcal{C}}_g, \tag{27}$$
$$\Delta\mathcal{R}_{\text{worst}}(g) \leq K_g \, \max_{\omega \in \Omega_{\text{worst}}} w_g(\omega)^2 \widehat{\mathcal{C}}_g. \tag{28}$$

Proposition 5 compares toggling one group while holding other components fixed. A full model contains multiple groups that may interact, so the proposition is used only to motivate an auditable diagnostic state rather than as a globally tight bound. The diagnostic is designed to be falsifiable through alignment counterfactuals that preserve marginal statistics while breaking sample-wise gate alignment.

We define an exposure-normalized pressure

$$\text{COP}_g = \frac{\widehat{\mathcal{C}}_g}{e_{\text{eff}}(g) + \epsilon}, \tag{29}$$

where $e_{\text{eff}}(g)$ is the EMA from Equation 16. TailPressure combines tail leverage and exposure-normalized strength:

$$\text{TP}_g = \ell_{\text{tail}}(g) \cdot \text{COP}_g. \tag{30}$$

Both $\text{COP}_g$ and $\text{TP}_g$ are computable from logged gates, parameter norms, and contract-fixed leverage coefficients.

## 6. H-CES: Auditable Closed-Loop Exposure Control

H-CES is a closed loop around the diagnostic state from Section 5. Figure 3 shows the three parts of the loop. First, the

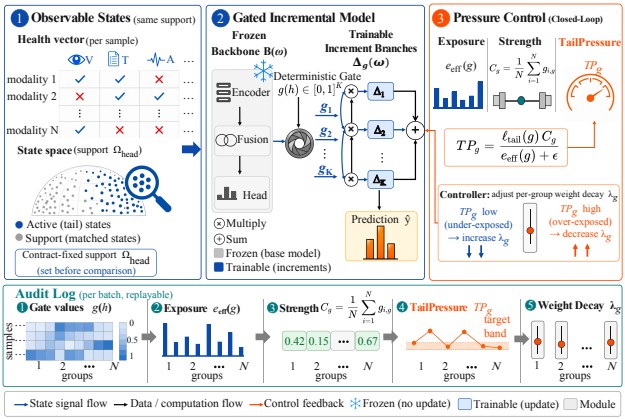

*Figure 3.* H-CES closed-loop and audit surface. The observable environment $\omega$ induces a deterministic gate that scales only increment branches. The controller reads exposure and strength statistics, computes TailPressure, and updates per-group decoupled weight decay. Each arrow corresponds to a logged quantity, so control decisions are replayable from logs.

observable environment $\omega$ determines a deterministic gate for each interaction group. Second, the training trace provides exposure and strength statistics on the same optimizer-step clock. Third, a per-group AdamW actuator changes only decoupled weight decay. The controller therefore acts on how fast each increment group can grow during training, rather than changing the task loss, adding an auxiliary branch, or modifying the inference call.

## 6.1. Replayable Gated Increments

At step $t$, each sample has an environment label $\omega_i = (b_i, q_i)$. The dependency map $\mathrm{Req}(g)$ and Equation 14 convert this label into $w_g(\omega_i)$. The forward pass then uses the decomposition in Equation 13: the frozen backbone is evaluated normally, and only the trainable increment branch of group $g$ is multiplied by $w_g(\omega_i)$. This design gives the controller an auditable surface. Given the batch environments and the fixed dependency map, every gate value can be recomputed after training, including the gates used under gradient accumulation and distributed data parallelism.

The gate is not a learned router and is not selected by validation labels. It is a deterministic measurement of whether the modalities required by group $g$ are present and reliable enough to contribute. As a result, H-CES does not create a new policy over environments. It makes an existing conditional interaction path measurable, so that exposure can be compared with the tail leverage fixed by the contract.

## 6.2. TailPressure Readout

H-CES reads the state less frequently than the optimizer updates. After every optimizer step, it all-reduces $\overline{w^2}_t(g)$ and updates $e_{\mathrm{eff}}(g)$ by Equation 16. At a controller tick,

it reads the current increment strength $\widehat{\mathcal{C}}_g$ from checkpoint norms, combines it with the exposure denominator in Equation 29, and multiplies by the contract-fixed leverage $\ell_{\mathrm{tail}}(g)$ in Equation 30. The state has a direct interpretation. A large value of $\mathrm{TP}_g$ means that a group with strong influence on the headline tail has accumulated more increment strength than its realized exposure can comfortably support.

Raw TailPressure values are not compared across unrelated routes. After warmup, H-CES collects a short history buffer for each eligible group and calibrates a group-specific center $\tau_g$ from the buffer median. The active band is

$$\tau_g^{\mathrm{low}} \leq \mathrm{TP}_g \leq \tau_g^{\mathrm{high}}, \quad \tau_g^{\mathrm{low}} = 0.5\tau_g, \quad \tau_g^{\mathrm{high}} = 2\tau_g. \tag{31}$$

This calibration prevents hand-written gate thresholds. The same route-level controller defaults are used across the matched-budget scan, and Table 6 checks that these defaults are not a narrow operating point.

## 6.3. Weight-Decay Actuation

The actuator is the decoupled weight decay $\lambda_g$ attached to the optimizer parameter group for $g$ (Loshchilov & Hutter, 2019). H-CES updates it only when $\mathrm{TP}_g$ leaves the calibrated band:

$$\lambda_g \leftarrow \mathrm{clip}\big(\lambda_g(\mathrm{TP}_g/\tau_g^{\mathrm{edge}})^\gamma, \lambda_{\min}, \lambda_{\max}\big), \tag{32}$$

where $\tau_g^{\mathrm{edge}} = \tau_g^{\mathrm{high}}$ for an upper-band event and $\tau_g^{\mathrm{edge}} = \tau_g^{\mathrm{low}}$ for a lower-band event.

This actuator is deliberately conservative. It does not resample examples, does not reweight the loss, and does not alter the gradient estimator. It only changes the decay term that controls increment growth after the gradient step. The trainable parameter count, forward graph, and inference-time computation remain those of the underlying increment interface. Algorithm 1 gives the optimizer-step order, including the DDP timing used to log $\overline{w^2}_t(g)$.

## 6.4. Static Baseline and Scope

Static-TP is the closest non-adaptive baseline. It uses the same contract leverage and exposure estimates, but assigns a fixed exposure-aware decay before training. H-CES differs in one respect: it closes the loop around the measured state during training. This distinction matters because the two parts of $\mathrm{TP}_g$ can drift in opposite directions. A group may remain covered by $\pi_{\mathrm{tr}}$ while its realized mini-batch exposure fluctuates, and its increment norm can grow faster than the available exposure. A static schedule can reflect the average contract, but it cannot react to this mismatch.

The method is not tied to LoRA. LoRA, adapters, prefix-style increments, and gated fusion blocks are all valid interfaces when they provide three objects: groups that map

*Table 2.* Full headline results under the locked same-support contract (score in Backbones are LLaVA-NeXT/Vicuna-7B (Chiang et al., 2023; Liu et al., 2023), Qwen3-VL-8B (Bai et al., 2025), and MulT (HME) (Tsai et al., 2019; Zhuang et al., 2025). External methods include LoRA (Hu et al., 2022), GroupDRO (Sagawa et al., 2020), global decoupled weight decay (Loshchilov & Hutter, 2019), DiGraP (Huang et al., 2025), LNLN (Zhang et al., 2024), and P-RMF (Zhu et al., 2025). Static-TP and H-CES are ours. All metrics are computed on the same $\Omega_{\text{head}}$, and each row is the single selected checkpoint from the same matched-budget rule.

| Backbone | Route | Method | Full | Train-Avg | Worst-Head | Shift-Reweight |
|---|---|---|---|---|---|---|
| LLaVA-NeXT Vicuna-7B | MMBench-EN | LoRA+GroupDRO | 66.6 | 61.4 | 48.7 | 53.0 |
| | | LoRA (global WD) | 67.4 | 61.2 | 45.9 | 50.8 |
| | | DiGraP | 67.2 | 61.3 | 49.4 | 53.6 |
| | | Static-TP (ours) | 67.0 | 61.3 | 50.4 | 54.2 |
| | | **H-CES** (ours) | 67.1 | 61.6 | **53.1** | **56.5** |
| | MMMU | LoRA+GroupDRO | 35.1 | 32.2 | 23.7 | 28.3 |
| | | LoRA (global WD) | 35.8 | 32.0 | 21.4 | 26.5 |
| | | DiGraP | 35.4 | 32.1 | 24.8 | 29.0 |
| | | Static-TP (ours) | 35.5 | 32.1 | 24.9 | 29.6 |
| | | **H-CES** (ours) | 35.6 | 32.3 | **27.2** | **31.0** |
| | MathVista | LoRA+GroupDRO | 34.0 | 31.0 | 22.1 | 26.3 |
| | | LoRA (global WD) | 34.6 | 30.8 | 19.8 | 24.6 |
| | | DiGraP | 34.2 | 30.9 | 22.9 | 27.0 |
| | | Static-TP (ours) | 34.3 | 30.9 | 23.5 | 27.7 |
| | | **H-CES** (ours) | 34.4 | 31.1 | **25.6** | **29.3** |
| Qwen3-VL-8B | MMBench-EN | LoRA+GroupDRO | 84.2 | 79.2 | 68.5 | 73.4 |
| | | LoRA (global WD) | 85.1 | 79.0 | 64.8 | 70.5 |
| | | DiGraP | 84.6 | 79.1 | 69.4 | 74.2 |
| | | Static-TP (ours) | 84.8 | 79.1 | 70.2 | 74.9 |
| | | **H-CES** (ours) | 84.9 | 79.3 | **72.9** | **77.1** |
| | MMMU | LoRA+GroupDRO | 66.5 | 62.2 | 50.9 | 57.1 |
| | | LoRA (global WD) | 67.4 | 62.0 | 47.1 | 53.6 |
| | | DiGraP | 66.8 | 62.1 | 52.1 | 58.1 |
| | | Static-TP (ours) | 67.0 | 62.1 | 52.8 | 58.7 |
| | | **H-CES** (ours) | 67.1 | 62.3 | **56.0** | **61.2** |
| | MathVista | LoRA+GroupDRO | 72.8 | 68.1 | 56.3 | 62.2 |
| | | LoRA (global WD) | 73.7 | 67.9 | 52.4 | 58.9 |
| | | DiGraP | 73.1 | 68.0 | 57.0 | 63.0 |
| | | Static-TP (ours) | 73.3 | 68.0 | 57.8 | 63.4 |
| | | **H-CES** (ours) | 73.4 | 68.2 | **60.5** | **65.9** |
| MulT (HME) | MOSI | LNLN | 84.9 | 79.1 | 69.2 | 72.7 |
| | | P-RMF | 84.8 | 79.2 | 70.6 | 73.8 |
| | | Static-TP (ours) | 84.7 | 79.1 | 71.9 | 75.0 |
| | | **H-CES** (ours) | 84.8 | 79.3 | **74.5** | **77.6** |
| | MOSEI | LNLN | 83.2 | 77.6 | 68.0 | 71.8 |
| | | P-RMF | 83.1 | 77.7 | 69.3 | 72.8 |
| | | Static-TP (ours) | 83.1 | 77.6 | 70.4 | 73.6 |
| | | **H-CES** (ours) | 83.2 | 77.8 | **72.9** | **75.8** |
| | CH-SIMS | LNLN | 81.4 | 76.0 | 65.9 | 69.4 |
| | | P-RMF | 81.3 | 76.0 | 67.1 | 70.4 |
| | | Static-TP (ours) | 81.3 | 75.9 | 68.5 | 71.2 |
| | | **H-CES** (ours) | 81.4 | 76.1 | **71.0** | **73.9** |

to optimizer parameter groups, a deterministic gate derived from observable environments, and a standard actuator on those groups during optimization. Section 7.5 evaluates adapter and prefix increments in the main text, and Appendix E.2 gives a non-PEFT MoE fusion example.

# 7. Experiments

## 7.1. Setup and Reporting Rules

We evaluate H-CES on two route families. The vision-language routes use MMBench-EN (dev v1.1), MMMU (val), and MathVista (mini) (Lu et al., 2024; Liu et al., 2024b; Yue et al., 2024) with LLaVA-NeXT/Vicuna-7B (Chiang et al., 2023; Liu et al., 2023) and Qwen3-VL-8B

(Bai et al., 2025). The sentiment routes use MOSI, MO-SEI, and CH-SIMS (Zadeh et al., 2017; Bagher Zadeh et al., 2018; Yu et al., 2020) with a MulT-style fusion backbone (Tsai et al., 2019). Each route uses the same write-once contract objects for missingness and degradation, and every headline metric is evaluated on the locked $\Omega_{\text{head}}$.

All methods follow the same matched-budget protocol. We train for the same number of optimizer steps, use the same scan budget, and select the checkpoint with the best Train-Avg subject to a fixed Full-drop constraint relative to the baseline. This rule prevents a method from buying tail gains by degrading the clean full-modality environment. The reporting unit is a method–route pair; reruns and controls reuse selected configurations rather than performing a

*Table 3.* Fixed-configuration five-seed robustness for Static-TP and H-CES on two representative LLaVA-NeXT/Vicuna-7B routes. Values are mean ± standard deviation.

| Route | Method | Full | Train-Avg | Worst-Head | Shift-Reweight |
|---|---|---|---|---|---|
| MMBench-EN | Static-TP | $66.9 \pm 0.1$ | $61.2 \pm 0.2$ | $50.1 \pm 0.7$ | $54.0 \pm 0.6$ |
| MMBench-EN | **H-CES** | $\mathbf{67.0 \pm 0.1}$ | $\mathbf{61.4 \pm 0.1}$ | $\mathbf{52.6 \pm 0.6}$ | $\mathbf{56.2 \pm 0.4}$ |
| MMMU | Static-TP | $35.5 \pm 0.2$ | $32.1 \pm 0.1$ | $24.6 \pm 0.5$ | $29.2 \pm 0.5$ |
| MMMU | **H-CES** | $\mathbf{35.6 \pm 0.1}$ | $\mathbf{32.2 \pm 0.1}$ | $\mathbf{26.7 \pm 0.5}$ | $\mathbf{30.8 \pm 0.4}$ |

*Table 4.* Alignment counterfactuals and same-information ablations (MMBench-EN dev v1.1, LLaVA-NeXT/Vicuna-7B; score in %; higher is better). All rows share the same $\Omega_{\text{head}}$ and the same selection rule. Results are single-run (1 seed).

| Method | Full | Worst-Head | Shift-Reweight |
|---|---|---|---|
| LoRA (tuned global WD) | **67.4** | 45.9 | 50.8 |
| Static-TP (no loop) | 67.0 | 50.4 | 54.2 |
| **H-CES (full)** | 67.1 | **53.1** | **56.5** |
| Train-gated only (train $w(\omega)$, eval $w \equiv 1$) | 67.1 | 49.1 | 53.3 |
| Eval-gated only (train $w \equiv 1$, eval $w(\omega)$) | 67.3 | 46.7 | 51.5 |
| shuffle-$w$ (train-time) | 67.1 | 46.2 | 51.0 |
| shuffle-$w$ (eval-time) | 67.1 | 46.0 | 50.9 |
| Random-gate (matched marginals) | 67.1 | 46.1 | 50.9 |
| Permuted controller mapping | 67.1 | 46.9 | 51.6 |
| Random grouping (same sizes) | 67.1 | 48.2 | 52.7 |
| Matched global WD schedule $\lambda(t)$ | 67.2 | 47.6 | 52.0 |
| Matched WD budget (uniform across groups) | 67.2 | 47.9 | 52.3 |
| Alt actuator: per-group LR scaling ($\propto 1/(e_{\text{eff}} + \epsilon)$) | 67.1 | 47.4 | 51.8 |

*Table 5.* Non-LoRA increment interfaces on the same backbone (MMBench-EN dev v1.1, LLaVA-NeXT/Vicuna-7B; score in %; higher is better). H-CES uses the same TailPressure state and the same selection rule as Table 2.

| Interface | Full | Train-Avg | Worst-Head | Shift-Reweight |
|---|---|---|---|---|
| Adapters (global WD) | 67.2 | 60.8 | 45.3 | 50.0 |
| Prefix (global WD) | 66.8 | 60.5 | 44.6 | 49.2 |
| Adapters + H-CES | 67.0 | 61.1 | **50.6** | **54.6** |
| Prefix + H-CES | 66.6 | 60.9 | **49.8** | **54.1** |

Shift-Reweight gains of 2.2 to 2.7 points. These changes are much larger than the corresponding Full changes, which remain within 0.2 points relative to Static-TP.

The method comparisons clarify where the gain comes from. GroupDRO directly optimizes environment groups, and DiGraP modifies gradient directions, yet both can leave the recurring adverse environments below the H-CES tail scores. Static-TP closes part of this gap, showing that exposure-aware regularization is useful even without feedback. H-CES then improves on Static-TP by responding to the measured TailPressure state during training, rather than relying on a fixed precomputed decay profile.

### 7.3. Fixed-Configuration Robustness

Table 3 reruns the selected Static-TP and H-CES configurations over five seeds. No per-seed re-selection is used, so the table measures training stochasticity rather than validation choice. On MMBench-EN, H-CES improves the mean Worst-Head and Shift-Reweight scores over Static-TP by 2.5 and 2.2 points. On MMMU, the corresponding gains are 2.1 and 1.6 points. The standard deviations are comparable across the two methods, indicating that the additional closed-loop gain is not driven by a single favorable seed under the fixed selected configurations.

### 7.4. Mechanism Checks and Same-Information Controls

The mechanism predicts that the tail gain should depend on gate-group-controller alignment. Table 4 tests this by breaking one part of the alignment while keeping the same $\Omega_{\text{head}}$, selection rule, and information budget. These rows are negative tests of the environment–group–actuator path: they pre-

second selection. Table 2 uses this single-checkpoint selection rule, while Table 3 reports fixed-configuration reruns for representative routes. Appendix E.1.1 gives multi-seed counterfactual checks under the same fixed configurations.

The vision-language baselines include LoRA with tuned global weight decay (Hu et al., 2022), LoRA+GroupDRO with environment groups (Sagawa et al., 2020), and Directional Gradient Projection (Huang et al., 2025). Static-TP is the closest direct baseline because it receives the same contract leverage and exposure information but does not close the loop. For sentiment routes, we compare LNLN (Zhang et al., 2024) and P-RMF (Zhu et al., 2025) under the same contract. Environment-stratified uncertainty, alternative fixed-support tail summaries, and score-level bridge checks are reported in Appendices C–C.3.

### 7.2. Headline Results on Same-Support Tails

Table 2 gives the complete headline comparison. Full and Train-Avg stay close within each route, so the separation is concentrated in Worst-Head and Shift-Reweight. Thus the table is a conditional tail comparison under preserved clean performance, not an unrestricted leaderboard. Across the six vision-language routes, H-CES improves over Static-TP by 2.1 to 3.2 points on Worst-Head and by 1.4 to 2.5 points on Shift-Reweight. The three sentiment routes show the same pattern, with Worst-Head gains of 2.5 to 2.6 points and

*Table 6.* One-dimensional sensitivity of H-CES controller defaults on MMBench-EN / LLaVA-NeXT-Vicuna-7B. Each row changes one quantity while keeping all other settings at the default.

| Setting | Full | Train-Avg | Worst-Head | Shift-Reweight |
|---|---|---|---|---|
| $e_{\mathrm{eff,min}} = 0.01$ | 67.1 | 61.6 | 52.8 | 56.2 |
| $e_{\mathrm{eff,min}} = 0.05$ | 67.0 | 61.4 | 52.3 | 55.7 |
| $\rho = 0.02$ | 67.1 | 61.5 | 52.6 | 56.0 |
| $\rho = 0.10$ | 67.0 | 61.5 | 52.4 | 55.8 |
| $K = 100$ | 67.0 | 61.6 | 52.9 | 56.3 |
| $K = 400$ | 67.1 | 61.4 | 52.2 | 55.6 |
| $\gamma = 0.3$ | 67.1 | 61.5 | 52.7 | 56.1 |
| $\gamma = 0.8$ | 67.0 | 61.5 | 52.5 | 55.9 |
| **Default** | **67.1** | **61.6** | **53.1** | **56.5** |

*Table 7.* Training overhead and audit footprint (MMBench-EN dev v1.1, LLaVA-NeXT/Vicuna-7B). Values are averages over 2000 post-warmup optimizer steps; trainable parameters are reported as a percentage of the backbone.

| Quantity | LoRA | H-CES |
|---|---|---|
| Trainable params | 0.20% | 0.20% |
| Step time (ms) | 338 | 346 |
| Throughput drop | 0.0% | 2.4% |
| Extra FLOPs | 0.0% | 0.9% |
| Peak memory (GB) | 29.2 | 29.8 |
| Log volume (MB / 1k steps) | 6.5 | 10.4 |
| Held-out diagnostic | N/A | $< 0.3\%$ |

serve the contract but break H-CES alignment. Train-gated only and eval-gated only do not recover the full improvement, so the result is not explained by gate placement alone. Shuffled gates preserve marginal gate statistics but destroy sample-wise alignment, and permuted mappings send controller signals to the wrong groups. Matched weight-decay budgets keep the overall regularization scale. In all cases, Full remains close, while the two tail scores move back toward the baselines under alignment breaks.

The same pattern appears in the diagnostic bridge. Figure 2(c) shows that TailPressure tracks headline tails inside the audited regime, and Figure 2(d) shows that preserving marginals while breaking alignment removes the gain. Appendix C.3 reports partial correlations, alternative summaries, and filtering robustness; Appendix E.1.1 repeats the core counterfactuals over five fixed-configuration seeds.

### 7.5. Interface Generality beyond LoRA

Table 5 replaces LoRA with adapter and prefix-style interfaces on the same LLaVA-NeXT/Vicuna-7B route (Houlsby et al., 2019; Li & Liang, 2021). The controller state, selection rule, and contract are unchanged. Because the route is fixed, the changed factor is the increment interface rather than dataset or contract. Adapters + H-CES improves Worst-Head from 45.3 to 50.6 and Shift-Reweight from 50.0 to 54.6; Prefix + H-CES gives a similar improvement. This supports the scope of H-CES: the method requires a groupable conditional interaction interface, not a particular low-rank parameterization. Appendix E.2 gives a non-PEFT MoE fusion example that uses the pressure state without LoRA.

### 7.6. Controller Defaults, Efficiency, and Boundaries

Table 6 varies one controller quantity at a time around the default setting. Worst-Head stays within 0.9 points and Shift-Reweight stays within 0.9 points of the default, while Full and Train-Avg remain nearly unchanged. The controller therefore does not rely on a route-specific sweet spot. This separates controller sensitivity, training cost, and the regime

where TailPressure is mechanism evidence.

Table 7 separates controller overhead from the held-out applicability diagnostic. Relative to the same LoRA interface, H-CES keeps the trainable parameter count at 0.20% of the backbone, increases step time from 338 ms to 346 ms, and increases peak memory from 29.2 GB to 29.8 GB. The added computation comes from gate statistics and periodic norm reads at training time, not from extra trainable branches or inference-time modules. The held-out diagnostic is outside the optimization state and adds less than 0.3% wall-clock time on the reference route.

TailPressure is interpreted only inside the audited regime. When exposure collapses for high-leverage groups or when applicability proxies exceed their thresholds, it becomes a boundary warning rather than mechanism evidence. Appendix E.5 gives the recognizable log signatures, and Appendix E.4 reports stress-only metrics on $\Omega_{\mathrm{stress}}$ without using them as headline claims.

## 8. Conclusion

This paper studied multimodal robustness under observable missingness and degradation with the evaluation support held fixed. Matching Full and Train-Avg on $\Omega_{\mathrm{head}}$ does not prevent large gaps in Worst-Head and Shift-Reweight. The evidence points to a coverage-exposure mismatch in conditional interaction, where environments are covered by the training mixture while high-leverage groups receive too little effective update signal. TailPressure makes this mismatch auditable from gate logs, contract leverage, exposure, and checkpoint strength. H-CES uses the same state to regulate per-group decoupled weight decay without changing the loss or inference interface. Across routes, backbones, increment interfaces, and strict alignment-breaking controls, H-CES improves same-support tail reliability while preserving clean performance. The boundary analyses further clarify that TailPressure should be interpreted only inside the audited regime, rather than universally.

## Impact Statement

This work provides tools for evaluating and improving multimodal reliability under observable missingness and degradation. The locked same-support contract can make robustness reports more comparable, and the replayable audit logs can help diagnose tail failures without adding inference-time branches. Potential risks include overfitting to a chosen environment discretization and exposing sensitive operational metadata through health signals or gate logs. We mitigate these risks by pre-registering contract objects, separating stress-only regimes from headline claims, and limiting H-CES to standard training-time regularization.

## Acknowledgments

This work was supported in part by the Guangdong Basic and Applied Basic Research Foundation (Nos. 2025A1515010225, 2025A1515060001), in part by the National Natural Science Foundation of China (No. 62302172), in part by the Youth Talent Development Program of the Guangdong Association for Science and Technology (No. SKXRC2025273), in part by the Key Area Project on Artificial Intelligence (Intelligent Robotics) under the Key Scientific Research Platform Program for Guangdong Higher Education Institutions (No. 2025ZDZX3043).

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

## A. Appendix Contents

# Appendix Guide

This appendix is an auditable companion to the main paper. It is organized so that the contract, controller, evidence checks, implementation details, extended experiments, boundary cases, and proofs can be located without following the floats sequentially.

**Appendix B: Algorithms, Contract Objects, and Notation.** Contains the optimizer-step H-CES pseudocode, notation table, object-role table, claim map, and the write-once SSOT objects for $\Omega_{\text{head}}$, $\Omega_{\text{worst}}$, $\pi_{\text{tr}}$, and $\pi_{\text{sh}}$. Use this part when reading Sections 3–6 or when checking whether a metric is evaluated on the same support.

**Appendix C: Phenomenon, Bridge Evidence, and Audit Rules.** Gives the pairing tolerances, all-pairs distributions, bootstrap uncertainty, per-environment stability, fixed-support tail-summary checks, TailPressure-score bridge, DDP timing, theory-to-log schema, applicability proxies, and TailPressure dynamics. This part supports Figure 2, Table 1, and the diagnostic claims in Sections 4–5.

**Appendix D: Implementation and Matched-Budget Details.** Defines Static-TP, explains same-information usage, lists grouping maps, records the controller defaults used by Table 6, and gives the matched-budget scan grids. Use this part when checking fairness of Table 2, Table 5, and Table 4.

**Appendix E: Extended Results and Stress Tests.** Collects appendix-only multi-seed counterfactuals, second-route counterfactual replication, marginal-gate checks, a non-PEFT example, overhead rules, stress-only metrics, boundary signatures, strength-proxy validation, and selection-rule details. The full route table and fixed-configuration seed robustness are now in Tables 2 and 3 in the main text.

**Appendix F: Proofs and Additional Derivations.** Provides the multi-group extension and complete derivations for Lemma 2 and Proposition 5. These derivations justify TailPressure as a surrogate-risk diagnostic, not as a direct bound on reported task scores.

# B. Algorithms, Contract Objects, and Notation

This section supports the formal contract and controller definitions in Sections 3–6. Algorithm 1 gives the replayable controller, Table 8 summarizes notation, and Tables 11–12 define the environment support used by the headline metrics.

## B.1. Algorithmic Details of H-CES

Algorithm 1 provides the complete, optimizer-step-aligned pseudocode needed to replay H-CES from logs. The DDP timing rules referenced by the algorithm are detailed in Appendix C.4.

---

**Algorithm 1** t-CES controller (optimizer-step aligned).

---

**Require:** Groups $\mathcal{G}$ mapped to optimizer parameter groups; per-group weight decay $\lambda_g$ (init $\lambda_{\text{init}}$).
**Require:** EMA rate $\rho$; controller period $K$; warmup $t_0$; exposure threshold $e_{\text{eff,min}}$; buffer length $L$.
**Require:** Stabilizer $\epsilon$; exponent $\gamma$; clipping bounds $\lambda_{\min}, \lambda_{\max}$; calibration factor $\kappa_g$.
 1: Initialize $e_{\text{eff}}(g) \leftarrow 0$ for all $g \in \mathcal{G}$; FIFO buffers $\mathcal{H}_g \leftarrow \emptyset$; targets $\tau_g \leftarrow$ unset.
 2: **for** optimizer steps $t = 1, 2, \ldots$ **do**
 3:     Build global batch $\mathcal{B}_t$ (after grad accumulation; across DDP workers; Appendix C.4).
 4:     Compute gates $w_g(\omega_i)$ for $i \in \mathcal{B}_t$ by Equation 14.
 5:     Forward by scaling only increment branches by $w_g(\omega_i)$; backbone unchanged.
 6:     Backward; update parameters with an optimizer step.
 7:     Compute $\overline{w^2}_t(g)$ by Equation 15 (DDP all-reduce; Appendix C.4).
 8:     Update $e_{\text{eff}}(g) \leftarrow (1 - \rho)e_{\text{eff}}(g) + \rho\,\overline{w^2}_t(g)$ by Equation 16.
 9:     **if** $t \bmod K = 0$ **then**
10:         **for** each $g \in \mathcal{G}$ **do**
11:             Compute $\widehat{\mathcal{C}}_g$ by Equation 21 and $\text{COP}_g$ by Equation 29.
12:             Compute leverage $\ell_{\text{tail}}(g)$ by Equation 22 and TailPressure $\text{TP}_g$ by Equation 30.
13:             Push $\text{TP}_g$ into FIFO $\mathcal{H}_g$ (keep last $L$).
14:             **if** $t \geq t_0$ **and** $e_{\text{eff}}(g) \geq e_{\text{eff,min}}$ **then**
15:                 **if** $\tau_g$ is unset **then**
16:                     Set $\tau_g \leftarrow \kappa_g \cdot \text{median}(\mathcal{H}_g)$.
17:                 **end if**
18:                 Set $\tau_g^{\text{low}} = 0.5\tau_g$ and $\tau_g^{\text{high}} = 2\tau_g$.
19:                 **if** $\text{TP}_g > \tau_g^{\text{high}}$ **then**
20:                     $\lambda_g \leftarrow \text{clip}\big(\lambda_g(\text{TP}_g/\tau_g^{\text{high}})^\gamma, \lambda_{\min}, \lambda_{\max}\big)$.
21:                 **else if** $\text{TP}_g < \tau_g^{\text{low}}$ **then**
22:                     $\lambda_g \leftarrow \text{clip}\big(\lambda_g(\text{TP}_g/\tau_g^{\text{low}})^\gamma, \lambda_{\min}, \lambda_{\max}\big)$.
23:                 **end if**
24:                 Write back $\lambda_g$ to the optimizer parameter group for $g$.
25:             **end if**
26:         **end for**
27:     **end if**
28: **end for**

---

## B.2. Notation and Naming Conventions

This appendix fixes naming conventions used throughout the paper. The main text uses a single name for each contract object and metric to avoid terminology drift.

We use Full for $\text{Score}_{\text{full}}$ in Equation 4. We use Train-Avg for $\text{Score}_{\text{tr-avg}}$ in Equation 5. We use Worst-Head for $\text{Score}_{\text{worst}}$ in Equation 6. We use Shift-Reweight for $\text{Score}_{\text{shift}}$ in Equation 7. All headline results and all mechanism evidence are on $\Omega_{\text{head}}$ unless explicitly marked as stress-only. The main text uses the term locked same-support contract, and contract refers to this object.

*Table 8.* SSOT notation used by the locked contract, diagnostics, and controller.

| Symbol | Meaning |
| --- | --- |
| $\omega = (b, q)$ | Observable environment signal with availability $b$ and quality $q$ (Equation 1). |
| $\Omega_{\text{test}}$ | Discrete set of observable environments enumerated by contract (Appendix B.4). |
| $\Omega_{\text{head}}$ | Headline same-support set used for all headline metrics and mechanism claims (Equation 2). |
| $\Omega_{\text{stress}}$ | Stress-only environments excluded from headline claims. |
| $\Omega_{\text{worst}}$ | Contract-fixed tail subset of $\Omega_{\text{head}}$ used by Worst-Head (Appendix B.4.3). |
| $\pi_{\text{tr}}, \pi_{\text{sh}}$ | Training mixture and shift reweighting distribution, both supported on $\Omega_{\text{head}}$. |
| $\text{Score}_\omega$ | Per-environment reported score in percent (Equation 3). |
| $\mathcal{R}_\omega$ | Per-environment surrogate risk for analysis (Equation 12). |
| $\mathcal{G}$ | Set of parameter groups associated with interaction modules (Equation 13). |
| $w_g(\omega)$ | Deterministic increment-branch gate for group $g$ (Equation 14). |
| $\overline{w^2}_t(g)$ | Step mean of $w_g(\omega)^2$ over the global batch (Equation 15). |
| $e_{\text{eff}}(g)$ | EMA exposure proxy computed from $\overline{w^2}_t(g)$ (Equation 16). |
| $\widehat{\mathcal{C}}_g$ | Strength proxy from increment-to-base norms (Equation 21). |
| $\text{COP}_g$ | Exposure-normalized pressure $\widehat{\mathcal{C}}_g/(e_{\text{eff}}(g) + \epsilon)$ (Equation 29). |
| $\ell_{\text{tail}}(g)$ | Tail leverage on fixed $\Omega_{\text{head}}$ and $\pi_{\text{sh}}$ (Equation 22). |
| $\text{TP}_g$ | TailPressure $= \ell_{\text{tail}}(g)\widehat{\mathcal{C}}_g/(e_{\text{eff}}(g) + \epsilon)$ (Equation 30). |
| $\tau_g^{\text{low}}, \tau_g^{\text{high}}$ | Controller band bounds for $\text{TP}_g$ (Equation 31). |
| $\lambda_g$ | Per-group decoupled weight decay actuator. |

*Table 9.* Roles and non-claims of the three objects used in the paper.

| Object | Role and scope boundary |
| --- | --- |
| Locked same-support contract | Fixes $\Omega_{\text{head}}$, $\Omega_{\text{worst}}$, $\pi_{\text{tr}}$, and $\pi_{\text{sh}}$ for comparable reporting. It is an evaluation protocol, not a replacement for worst-group risk, CVaR, or other robust objectives. |
| TailPressure | Measures one gate-aligned source of same-support tail instability from logged gates, tail leverage, exposure, and strength proxies. It is a partial auditable diagnostic, not a complete explanation of tail variance or a calibrated harm predictor. |
| H-CES | Uses TailPressure to adjust per-group decoupled weight decay during training. It does not change the task loss, add inference branches, or introduce new trainable parameters beyond the underlying increment interface. |

## B.3. Object Roles and Claim Map

This subsection fixes the scope of the three named objects used in the paper. The main text uses a short description to avoid duplicating terminology tables, while Table 9 gives the full role separation and Table 10 maps each claim to the corresponding evidence.

## B.4. Single Source of Truth for the Contract Objects

This appendix fixes all contract objects as a single source of truth. For each route, all methods share:

1. discrete environment support $\Omega_{\text{test}}$ and headline subset $\Omega_{\text{head}}$;

2. task-valid predicate $\text{Valid}(\omega)$ defining which missingness patterns enter $\Omega_{\text{head}}$;

3. contract-fixed tail subset $\Omega_{\text{worst}} \subseteq \Omega_{\text{head}}$;

4. deterministic quality bucket mapping to $q \in \{1.0, 0.7, 0.4, 0.2, 0.0\}$;

5. training mixture $\pi_{\text{tr}}$ supported on $\Omega_{\text{head}}$ and deterministic corruption realizations;

6. shift family $\pi_{\text{sh}}$ supported on the same $\Omega_{\text{head}}$ and its fixed grid.

*Table 10.* Claim map connecting the paper claims to their evidence and scope.

| Claim | Evidence and intended scope |
|---|---|
| $w^2$ enters group gradients and local curvature under gated increments. | Lemma 2; twice differentiability is used only for the gradient and curvature scaling statement. |
| Single-group surrogate-risk sensitivity carries tail leverage on fixed $\Omega_{\text{head}}$. | Proposition 5; $\beta$-smoothness is used for the increment inequality. This is a surrogate-risk statement, not a theorem about reported task scores. |
| TailPressure should combine $w^2$, $\ell_{\text{tail}}$, strength, and exposure normalization. | Equations 15–30 and the audit schema in Table 18; this motivates an auditable diagnostic state. |
| Same-support tail failure exists despite matched Full and Train-Avg. | Section 4, Table 1, and Figure 2; all aggregates are evaluated on the locked $\Omega_{\text{head}}$. |
| TailPressure is related to reported Worst-Head and Shift-Reweight scores only through empirical bridge evidence. | Appendix C.3, Table 15, Table 16, Table 17, and Figure 8; interpretation is restricted to the audited regime. |
| The additional gain of H-CES comes from aligned closed-loop regulation rather than matched information alone. | Table 4 and Appendix E.1.1; alignment-breaking counterfactuals preserve Full but remove most of the tail gain. |
| Severe missingness gives recognizable boundaries for interpreting TailPressure. | Appendix E.5, Table 26, and Figure 10; outside the audited regime, TailPressure is used as a warning signal rather than mechanism evidence. |

*Table 11.* SSOT: discrete environment support $\Omega_{\text{test}}$ for vision–language routes (modalities: image I, text T). Rows marked stress belong to $\Omega_{\text{stress}}$ and are excluded from $\Omega_{\text{head}}$.

| Environment | Availability $b$ | Quality $q$ |
|---|---|---|
| Full | (I=1, T=1) | (1.0, 1.0) |
| Hard-missing I (headline) | (0, 1) | (0.0, 1.0) |
| Hard-missing T (stress; task-invalid) | (1, 0) | (1.0, 0.0) |
| Degrade I $q = 0.7$ | (1, 1) | (0.7, 1.0) |
| Degrade I $q = 0.4$ | (1, 1) | (0.4, 1.0) |
| Degrade I $q = 0.2$ | (1, 1) | (0.2, 1.0) |
| Degrade I $q = 0.0$ (stress) | (1, 1) | (0.0, 1.0) |
| Degrade T $q = 0.7$ | (1, 1) | (1.0, 0.7) |
| Degrade T $q = 0.4$ | (1, 1) | (1.0, 0.4) |
| Degrade T $q = 0.2$ | (1, 1) | (1.0, 0.2) |
| Degrade T $q = 0.0$ (stress) | (1, 1) | (1.0, 0.0) |
| Joint degrade $q = (0.4, 0.4)$ | (1, 1) | (0.4, 0.4) |
| Joint degrade $q = (0.2, 0.4)$ | (1, 1) | (0.2, 0.4) |
| Joint degrade $q = (0.4, 0.2)$ | (1, 1) | (0.4, 0.2) |

### B.4.1. DISCRETE ENVIRONMENT SUPPORT TABLES

Table 11 defines $\Omega_{\text{test}}$ for image–text routes. Image dropout is treated as task-valid and included in $\Omega_{\text{head}}$. Missing text is task-invalid and placed in $\Omega_{\text{stress}}$.

For sentiment routes, text is always present and the protocol applies to audio and vision. Hard-missing audio or vision is task-valid and included in $\Omega_{\text{head}}$. Table 12 defines $\Omega_{\text{test}}$.

### B.4.2. TASK-VALIDITY PREDICATE

For vision–language, $\text{Valid}(\omega) = 1$ if and only if text is available. Image dropout is treated as task-valid because labels and the score function remain defined. For MSA, $\text{Valid}(\omega) = 1$ for all environments in Table 12 except those explicitly marked stress.

*Table 12.* SSOT: discrete environment support $\Omega_{\text{test}}$ for the MSA sub-protocol (modalities: audio A, vision V; text always present). Rows marked stress belong to $\Omega_{\text{stress}}$ and are excluded from $\Omega_{\text{head}}$.

| Environment | Availability $b$ | Quality $q$ |
|---|---|---|
| Full | (A=1, V=1) | (1.0, 1.0) |
| Hard-missing A (headline) | (0, 1) | (0.0, 1.0) |
| Hard-missing V (headline) | (1, 0) | (1.0, 0.0) |
| Degrade A $q = 0.7$ | (1, 1) | (0.7, 1.0) |
| Degrade A $q = 0.4$ | (1, 1) | (0.4, 1.0) |
| Degrade A $q = 0.2$ | (1, 1) | (0.2, 1.0) |
| Degrade A $q = 0.0$ (stress) | (1, 1) | (0.0, 1.0) |
| Degrade V $q = 0.7$ | (1, 1) | (1.0, 0.7) |
| Degrade V $q = 0.4$ | (1, 1) | (1.0, 0.4) |
| Degrade V $q = 0.2$ | (1, 1) | (1.0, 0.2) |
| Degrade V $q = 0.0$ (stress) | (1, 1) | (1.0, 0.0) |
| Joint degrade $q = (0.4, 0.4)$ | (1, 1) | (0.4, 0.4) |
| Joint degrade $q = (0.2, 0.4)$ | (1, 1) | (0.2, 0.4) |
| Joint degrade $q = (0.4, 0.2)$ | (1, 1) | (0.4, 0.2) |

### B.4.3. TAIL SUBSET $\Omega_{\text{worst}}$

We fix $\Omega_{\text{worst}}$ by a deterministic rule to avoid post-hoc tail selection. Define severity score

$$s_{\text{sev}}(\omega) = \sum_m (1 - q_m(\omega)) + \eta \sum_m \mathbf{1}_{\{b_m(\omega)=0\}}, \tag{33}$$

with $\eta = 1$ fixed per route. Let $K_{\text{worst}} = 4$ for two-modality supports. Then $\Omega_{\text{worst}}$ contains the $K_{\text{worst}}$ environments in $\Omega_{\text{head}}$ with the largest $s_{\text{sev}}(\omega)$, breaking ties by lexicographic order on $(b, q)$.

### B.4.4. SHIFT FAMILY AND GRID

The shift distribution is a deterministic reweighting over the same $\Omega_{\text{head}}$:

$$\pi_{\text{sh}}^{(\alpha,\beta)}(\omega) \propto \pi_{\text{ref}}(\omega) \cdot \exp\left(\alpha \cdot \text{sev}(\omega) + \beta \cdot \#\text{miss}(\omega)\right), \tag{34}$$

where $\text{sev}(\omega) = \sum_m (1 - q_m(\omega))$ and $\#\text{miss}(\omega) = \sum_m \mathbf{1}_{\{b_m(\omega)=0\}}$. We fix

$$\alpha \in \{0, 1, 2\}, \qquad \beta \in \{0, 1\}, \tag{35}$$

and use $\pi_{\text{ref}} = \text{Unif}(\Omega_{\text{head}})$ unless stated otherwise.

### B.4.5. TRAINING MIXTURE

The training mixture is supported on $\Omega_{\text{head}}$:

$$\pi_{\text{tr}} = p_{\text{full}}\delta_{\omega_{\text{full}}} + (1 - p_{\text{full}}) \text{Unif}(\Omega_{\text{head}} \setminus \{\omega_{\text{full}}\}). \tag{36}$$

Unless stated, $p_{\text{full}} = 0.6$.

### B.4.6. SENSITIVITY TO DISCRETIZATION AND SHIFT-FAMILY VARIANTS

We vary contract objects within a pre-registered family that preserves same-support. We change the number of quality buckets, the shift grid in Equation 35, and the tail size $K_{\text{worst}}$. Across these variants, the ordering between H-CES and strong static baselines remains consistent on headline tails, and the correlation between $\max_g \text{TP}_g$ and Worst-Head remains stable within the audited regime. Figure 4 summarizes the trend on a reference route, and Table 13 gives the corresponding numerical values.

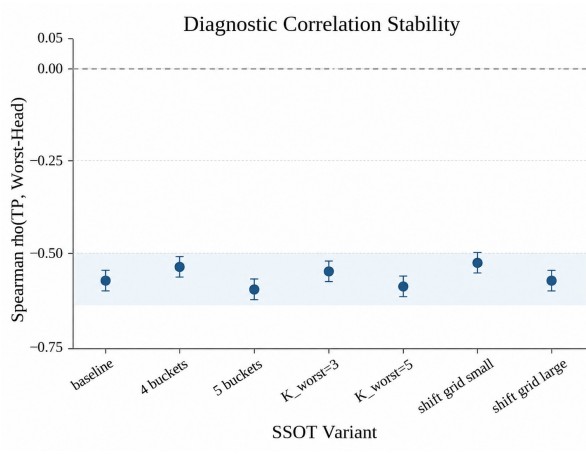

(a) TailPressure Correlation Across Contract Variants

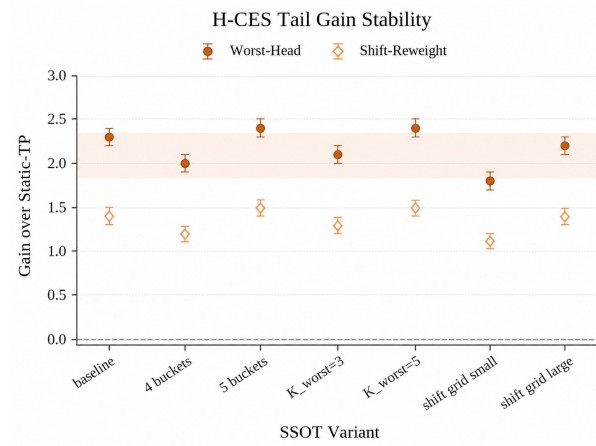

(b) H-CES Tail Gain Across Contract Variants

*Figure 4.* Sensitivity of the diagnostic and the controller to SSOT variants. (a) Spearman correlation between post-warmup $\max_g \mathrm{TP}_g$ and Worst-Head across scan points within the audited regime under each SSOT variant. (b) Worst-Head gain of H-CES over Static-TP under the same Full-drop constraint for the same variants. Each marker corresponds to one discretization or shift-grid variant that preserves the same-support property.

*Table 13.* Sensitivity to SSOT variants on the reference route. All variants preserve the locked same-support property and change only one pre-registered contract choice.

| SSOT variant | $\rho(\max_g \mathrm{TP}_g, \text{Worst-Head})$ | $\Delta$ Worst-Head | $\Delta$ Shift-Reweight |
|---|---|---|---|
| baseline | $-0.57$ | $+2.3$ | $+1.4$ |
| 4 buckets | $-0.54$ | $+2.0$ | $+1.2$ |
| 5 buckets | $-0.60$ | $+2.4$ | $+1.5$ |
| $K_{\text{worst}} = 3$ | $-0.55$ | $+2.1$ | $+1.3$ |
| $K_{\text{worst}} = 5$ | $-0.59$ | $+2.4$ | $+1.5$ |
| shift grid small | $-0.52$ | $+1.8$ | $+1.1$ |
| shift grid large | $-0.56$ | $+2.2$ | $+1.4$ |

# C. Phenomenon, Bridge Evidence, and Audit Rules

This section provides the evidence trail behind the phenomenon and diagnostic claims. Figure 5 and Table 1 support the same-support pairing analysis, Figure 6 checks whether the adverse subset is stable, Figure 7 checks alternative fixed-support tail summaries, and Tables 15–17 plus Figure 8 document the TailPressure-score bridge.

## C.1. Phenomenon Validity and Uncertainty

### C.1.1. PAIRING TOLERANCES

We fix tolerances in Equation 8 to

$$\delta_{\text{full}} = 0.2, \qquad \delta_{\text{avg}} = 0.2, \tag{37}$$

in score points.

### C.1.2. PAIRING ALGORITHM AND ALL-PAIRS DISTRIBUTIONS

Figure 5 shows the all-pairs gap distributions used to summarize Table 1.

Given a run set $\mathcal{R}$ with recorded $(\text{Score}_{\text{full}}, \text{Score}_{\text{tr-avg}})$, define the eligible set

$$\mathcal{P}_{\text{elig}} = \{(a, b) \in \mathcal{R} \times \mathcal{R} : a < b, \text{ Equation 8 holds}\}. \tag{38}$$

We report quantiles and histograms of $\Delta_{\text{worst}}$ and $\Delta_{\text{shift}}$ over all eligible pairs without sub-selection. To avoid cherry-picking

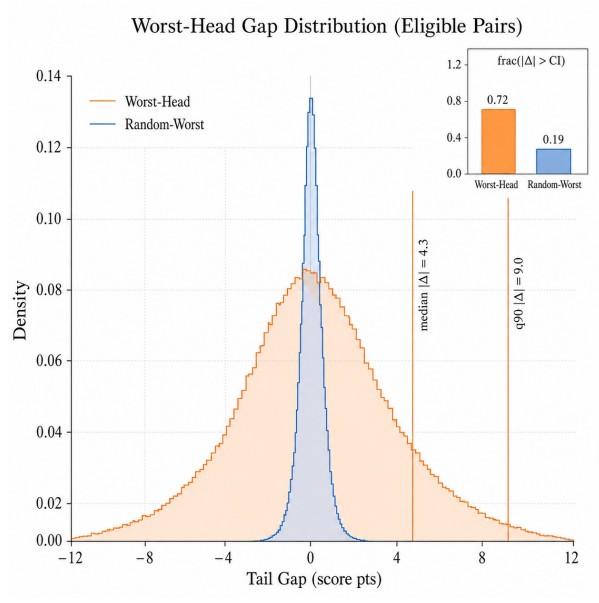

(a) Matched-Pair Gaps in Worst-Head Scores

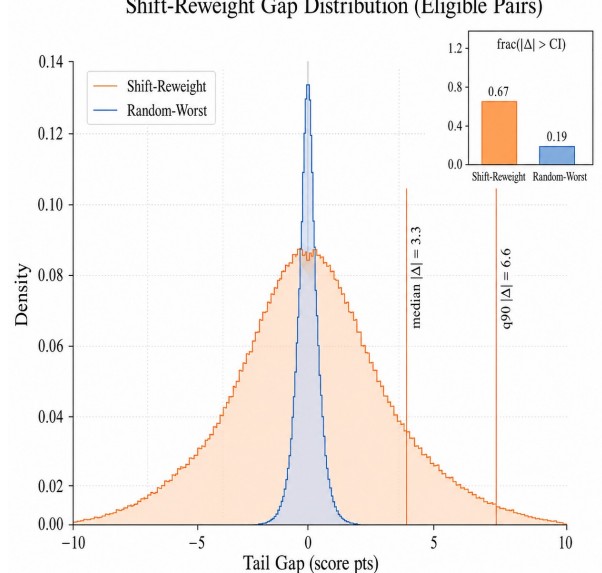

(b) Matched-Pair Gaps in Shift-Reweight Scores

*Figure 5.* All eligible pairs under the write-once tolerances (pairs match Full and Train-Avg). (a) Histogram of Worst-Head gaps $\Delta_{\text{worst}}$ over $\mathcal{P}_{\text{elig}}$, with the Random-Worst same-support control overlaid for comparison. (b) Histogram of Shift-Reweight gaps $\Delta_{\text{shift}}$, again with Random-Worst overlaid; the inset reports the fraction of eligible pairs whose $|\Delta|$ exceeds bootstrap CI widths for Worst-Head, Shift-Reweight, and Random-Worst.

*Table 14.* Per-environment evaluation sample counts on $\Omega_{\text{head}}$ (representative; used to interpret Worst-Head uncertainty). Counts are from the fixed evaluation split after assigning each example to an environment by its observable $(b, q)$.

| Route | min $n_\omega$ | q10 | q50 | q90 |
|---|---|---|---|---|
| MMBench-EN (dev v1.1) | 320 | 410 | 560 | 870 |
| MMMU (val) | 180 | 240 | 330 | 520 |
| MOSI | 230 | 300 | 420 | 610 |

in visualizations, Figure 2(a) uses a deterministic representative pair. We sort $\mathcal{P}_{\text{elig}}$ by $|\Delta_{\text{shift}}|$ with ties broken by run id and choose the median element.

### C.1.3. PER-ENVIRONMENT SAMPLE COUNTS

Let $n_\omega$ be the number of test samples assigned to environment $\omega \in \Omega_{\text{head}}$. The quantiles in Table 14 guard against interpretations driven by tiny environments.

### C.1.4. BOOTSTRAP UNCERTAINTY FOR WORST-HEAD

We compute an environment-stratified bootstrap CI for $\text{Score}_{\text{worst}}$. For each $\omega \in \Omega_{\text{worst}}$, we resample test examples within $\omega$ with replacement, recompute $\text{Score}_\omega$, take $\min_{\omega \in \Omega_{\text{worst}}}$, and repeat $B = 2000$ times to obtain percentile intervals. We report 95% intervals and quantify how often eligible-pair tail gaps exceed CI widths.

### C.1.5. PER-ENVIRONMENT MATRIX STABILITY

Figure 6 visualizes whether the recurring adverse environments in Figure 2(b) remain stable across eligible pairs.

Let $A \in \mathbb{R}^{|\Omega_{\text{head}}|}$ collect per-environment scores $\text{Score}_\omega$ for a run. We report rank correlations between runs and the concentration of tail gaps on a recurring subset of environments.

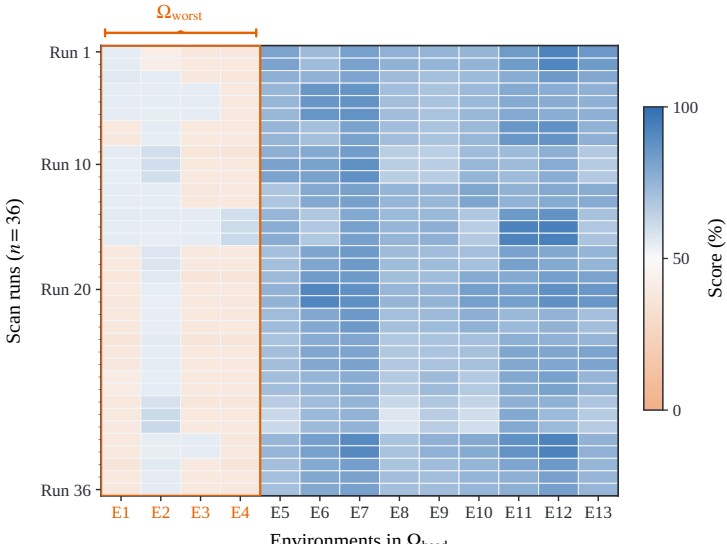

*Figure 6.* Per-environment score matrix stability on $\Omega_{\mathrm{head}}$. Heatmap shows per-environment scores $\mathrm{Score}_\omega$ (in %) for each scan run (rows) across environments in $\Omega_{\mathrm{head}}$ (columns), with the contract-defined $\Omega_{\mathrm{worst}}$ bracketed. The recurring low-score columns indicate a stable subset of adverse environments inside the fixed same-support set.

### C.2. Tail Summaries on the Fixed Support

Figure 7(a), Figure 7(b), and Figure 7(c) compare fixed-support tail summaries that use the same $\Omega_{\mathrm{head}}$ but vary the tail functional.

Let $\{\widehat{\mathrm{Score}}_\omega(\theta) : \omega \in \Omega_{\mathrm{head}}\}$ denote empirical per-environment scores (in percent) as in Equation 3. Let $\widehat{\mathrm{Score}}_{[1]} \leq \widehat{\mathrm{Score}}_{[2]} \leq \cdots$ denote the sorted scores in ascending order (lower is worse). A same-support worst-$k$ score summary is

$$\widehat{\mathrm{Score}}_{\mathrm{worst\text{-}}k}(\theta) = \frac{1}{k} \sum_{j=1}^{k} \widehat{\mathrm{Score}}_{[j]}(\theta). \tag{39}$$

A CVaR-like score tail summary at level $\alpha \in (0, 1]$ is the average of the lowest $\lceil \alpha|\Omega_{\mathrm{head}}| \rceil$ environments:

$$\widehat{\mathrm{Score}}_{\mathrm{cvar\text{-}}\alpha}(\theta) = \frac{1}{\lceil \alpha|\Omega_{\mathrm{head}}| \rceil} \sum_{j=1}^{\lceil \alpha|\Omega_{\mathrm{head}}| \rceil} \widehat{\mathrm{Score}}_{[j]}(\theta). \tag{40}$$

These summaries are computed on the same fixed support $\Omega_{\mathrm{head}}$ and provide a consistency check that H-CES improvements are not specific to the single-environment minimum used by Worst-Head.

### C.3. Metric Alignment Bridge

Table 15, Table 16, Table 17, and Figure 8 provide the empirical bridge between the auditable TailPressure state and the reported score-level tail metrics.

This appendix provides the bridge between the surrogate-risk analysis and the reported score tails. The paper does not claim a formal implication from $\mathcal{R}$ to Score. Instead, we test a mechanism prediction within an audited regime. TailPressure should co-vary with score-tail outcomes, and alignment-breaking counterfactuals should remove both the tail gains and the diagnostic correlation.

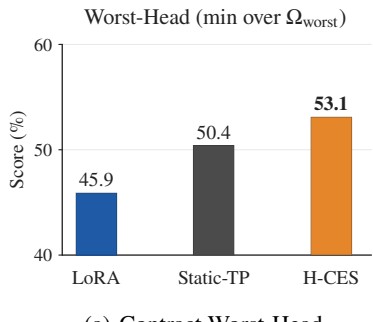 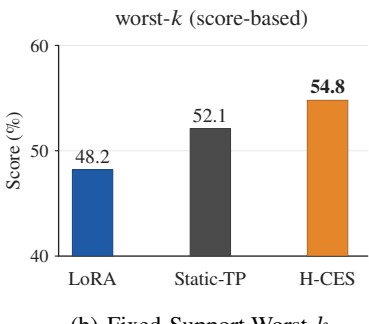 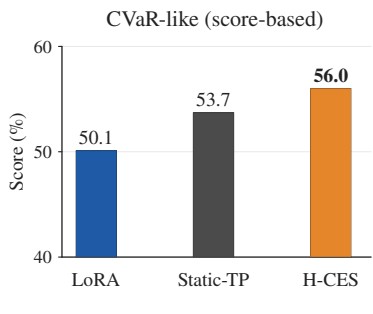

(a) Contract Worst-Head        (b) Fixed-Support Worst-$k$        (c) Fixed-Support CVaR Tail

*Figure 7.* Same-support tail-summary consistency on the reference route. (a) Worst-Head uses the contract-defined $\Omega_{\text{worst}}$. (b) worst-$k$ averages the lowest $k$ empirical per-environment scores on $\Omega_{\text{head}}$. (c) The CVaR-like score tail averages the lowest fraction of environments on the same fixed support. H-CES improves all three summaries, showing that the gain is not specific to a single-environment minimum.

*Table 15.* Bridge summary (MMBench-EN dev v1.1, LLaVA-NeXT/Vicuna-7B). Values are Spearman $\rho$ between $T(r)$ and score tails, computed on in-regime scan runs. Within-bin is the median correlation across Full and Train-Avg bins. Partial controls for Full, Train-Avg, and step count. Counterfactual rows use the same scan grid but break alignment.

| Setting | $\rho(T, \text{Worst-Head})$ | $\rho(T, \text{Shift-Reweight})$ |
|---|---|---|
| In-regime (global) | $-0.69$ | $-0.63$ |
| Within-bin median | $-0.57$ | $-0.51$ |
| Partial (Full, Train-Avg, steps) | $-0.49$ | $-0.45$ |
| shuffle-$w$ (train-time) | $-0.07$ | $-0.05$ |
| Permuted controller mapping | $-0.10$ | $-0.08$ |

### C.3.1. BRIDGE PROTOCOL

For each route and method, collect a run set $\mathcal{R}$ from a matched-budget scan. For each run $r \in \mathcal{R}$, compute

$$T(r) = \max_{g} \text{TP}_g(r) \quad \text{(post-warmup average)}, \tag{41}$$

$$S_{\text{worst}}(r) = \text{Score}_{\text{worst}}(r), \tag{42}$$

$$S_{\text{shift}}(r) = \text{Score}_{\text{shift}}(r). \tag{43}$$

We report Pearson and Spearman correlations with bootstrap confidence intervals over runs. We exclude out-of-regime points only via the SSOT applicability thresholds in Table 19.

### C.3.2. EVIDENCE STRUCTURE

We stratify runs by Full and Train-Avg into bins of width 0.5 score points and compute within-bin Spearman correlations to reduce confounding by global training quality. We also report partial correlations that control for Full, Train-Avg, and training step count at evaluation time through linear residualization. Finally, we repeat the same correlation analysis under shuffle-$w$ and permuted mapping counterfactuals that preserve marginal statistics while breaking alignment.

Figure 8(a) and Figure 8(b) plot the reference-route bridge for Worst-Head and Shift-Reweight, respectively.

### C.4. DDP Alignment and Controller Timing

Let worker $r$ hold local batch $\mathcal{B}_t^{(r)}$ at optimizer step $t$. Compute local sums $s_{t,g}^{(r)} = \sum_{i \in \mathcal{B}_t^{(r)}} w_g(\omega_i)^2$ and local counts $B^{(r)} = |\mathcal{B}_t^{(r)}|$. All-reduce

$$s_{t,g} = \sum_r s_{t,g}^{(r)}, \qquad B_{\text{glob}} = \sum_r B^{(r)}. \tag{44}$$

*Table 16.* Auditable diagnostic alternatives on the reference route. Values summarize association with score tails within the audited regime. TailPressure is the strongest tested summary but remains a partial diagnostic rather than a complete explanation of tail variance.

| Summary statistic | $\rho(T, \text{Worst})$ | partial $\rho$ | $\Delta R^2$ | $\rho(T, \text{Shift})$ |
|---|---|---|---|---|
| $\max_g \text{TP}_g$ | $-0.69$ | $-0.49$ | **0.12** | $-0.63$ |
| $\text{mean}_g \text{TP}_g$ | $-0.61$ | $-0.42$ | 0.08 | $-0.56$ |
| $\max_g \ell_{\text{tail}}(g)\widehat{\mathcal{C}}_g/(\mathbb{E}[w_g] + \epsilon)$ | $-0.59$ | $-0.40$ | 0.07 | $-0.54$ |
| $\max_g \text{COP}_g$ | $-0.55$ | $-0.36$ | 0.06 | $-0.49$ |
| $\max_g \ell_{\text{tail}}(g)\widehat{\mathcal{C}}_g$ | $-0.46$ | $-0.28$ | 0.03 | $-0.41$ |
| $\max_g \widehat{\mathcal{C}}_g$ | $-0.33$ | $-0.19$ | 0.02 | $-0.30$ |

*Table 17.* Bridge-filtering robustness. Out-of-regime exclusion is used only for mechanism-bridge analysis, not for headline method comparison. The denominator is the scan-run pool for the bridge analysis.

| Route | Current exclusion | $\rho_{\text{worst}}$ (all / current / soft) | $\rho_{\text{shift}}$ (all / current / soft) |
|---|---|---|---|
| MMBench-EN / LLaVA-NeXT | 9.4% (17/180) | $-0.58 / -0.69 / -0.64$ | $-0.53 / -0.63 / -0.59$ |
| MMMU / LLaVA-NeXT | 12.2% (22/180) | $-0.48 / -0.57 / -0.53$ | $-0.43 / -0.50 / -0.47$ |
| MOSI / MulT (HME) | 6.3% (9/144) | $-0.46 / -0.54 / -0.51$ | $-0.42 / -0.49 / -0.46$ |

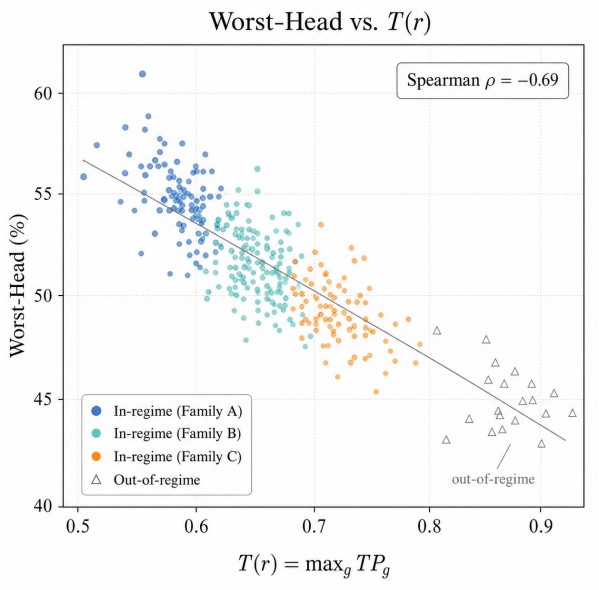

(a) TailPressure Versus Worst-Head Score

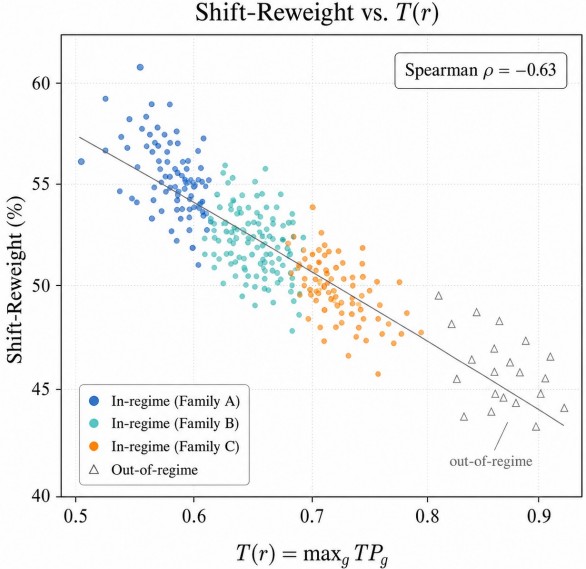

(b) TailPressure Versus Shift-Reweight Score

*Figure 8.* Bridge plots: TailPressure versus score tails. (a) shows $T(r) = \max_g \text{TP}_g(r)$ versus Worst-Head, and (b) shows $T(r)$ versus Shift-Reweight across scan points. Points flagged as out-of-regime by applicability proxies are shown with a distinct marker and excluded only from mechanism-bridge correlation estimates, not from headline method comparisons.

*Table 18.* Audit map linking theory symbols to logged fields and timing rules. The map is replayable without hidden optimizer state.

| Symbol | Meaning | Logged field and timing rule |
|---|---|---|
| $w_g(\omega)$ | gate for group $g$ | `gate/w_quantiles_g` (p1/p10/p50/p90/p99), `gate/w_zero_rate_g` (post-forward) |
| $\overline{w^2}_t(g)$ | step mean($w^2$) | `exposure/w2_mean_step_g` (post-step, after all-reduce) |
| $e_{\text{eff}}(g)$ | EMA exposure rate | `exposure/eeff_ema_g` (post-step) |
| $\widehat{\mathcal{C}}_g$ | strength proxy | `strength/cplx_g` (controller tick, fp32) |
| $\text{COP}_g$ | exposure-normalized pressure | `pressure/cop_g` (controller tick) |
| $\ell_{\text{tail}}(g)$ | tail leverage | `contract/tail_leverage_g` (precomputed once per route) |
| $\text{TP}_g$ | TailPressure | `pressure/tailpressure_g` (controller tick) |
| $\tau_g^{\text{low}}, \tau_g^{\text{high}}$ | band bounds | `target/tau_low_g`, `target/tau_high_g` (controller tick) |
| $\lambda_g$ | per-group weight decay | `actuator/wd_g`, `actuator/wd_clip_event_g` (controller tick) |

*Table 19.* Auditable proxies for applicability of Proposition 5. Thresholds define an audited regime used for mechanism evidence.

| Condition | Proxy (computed on held-out batches) | Threshold rule (audited regime) |
|---|---|---|
| Energy bridge in Equation 23 | $r_{\text{bridge}} = \dfrac{\|\Delta z_g\|_2^2}{w_g(\omega)^2 \widehat{\mathcal{C}}_g + \delta}$ | $q_{0.95}(r_{\text{bridge}}) \leq \bar{r}_{\text{bridge}}$ |
| First-order residual in Equation 24 | $r_1 = \dfrac{|\langle \nabla_z \ell(z_{\text{base}}), \Delta z_g \rangle|}{\|\Delta z_g\|_2^2 + \delta}$ | $q_{0.95}(r_1) \leq \bar{r}_1$ |

Then

$$\overline{w^2}_t(g) = \frac{s_{t,g}}{B_{\text{glob}}}. \tag{45}$$

This statistic matches the single-process mean when an optimizer step is defined after gradient accumulation and before the parameter update.

### C.4.1. Time-Axis Checkpoints

We fix a 5-checkpoint time axis. Pre-forward assembles the global batch and computes $w_g(\omega_i)$. Post-forward computes losses. Post-backward finishes gradients. Post-step applies the optimizer update and updates $e_{\text{eff}}$. Controller tick runs every $K$ steps, computes $(\widehat{\mathcal{C}}_g, \text{TP}_g)$, and may update $\lambda_g$.

## C.5. Audit Map and Applicability Criteria

### C.5.1. Theory-to-Log Schema

Table 18 lists the logged fields used to replay the diagnostic state and controller decisions. The table also fixes when each field is written, which is necessary for matching the optimizer-step exposure definition in Equation 15.

### C.5.2. Applicability Proxies and Thresholds

Conditions 3 and 4 are treated as auditable applicability conditions rather than as untestable assumptions. We estimate proxies on held-out batches every $K_{\text{diag}}$ steps and log quantiles per group.

We use $\delta = 10^{-12}$, $K_{\text{diag}} = 500$ steps, and conservative thresholds $\bar{r}_{\text{bridge}} = 50$ and $\bar{r}_1 = 0.15$, fixed per route before method comparisons.

### C.5.3. How We Compute $\Delta z_g$ for Proxies

Computing $\Delta z_g$ naively by ablating each group per step would be expensive. We use a periodic held-out diagnostic that is decoupled from the training loss. Every $K_{\text{diag}}$ steps, we sample one held-out micro-batch and run two forward passes in fp16 with shared activations. The first pass is a normal forward with all groups on. The second pass masks one target group by setting its increment gate to zero for that pass only. We record $\Delta z_g$ at the loss input of the model head. We rotate over

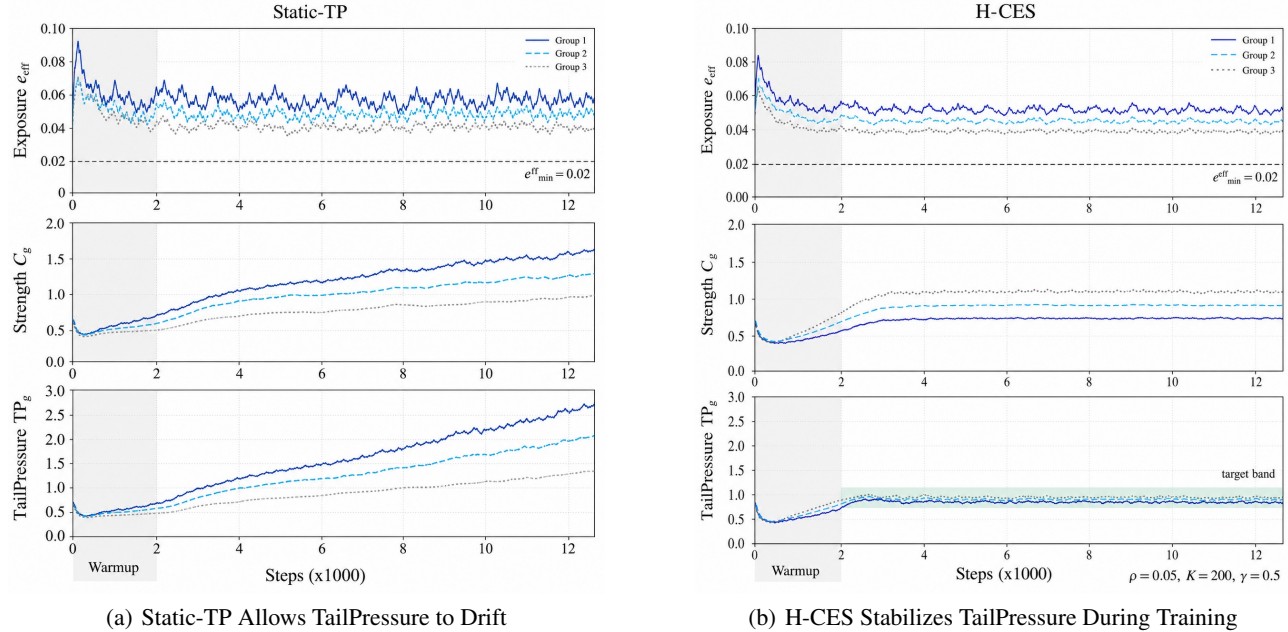

(a) Static-TP Allows TailPressure to Drift                    (b) H-CES Stabilizes TailPressure During Training

*Figure 9.* TailPressure non-stationarity under a fixed mixture. (a) Static-TP exposure $e_{\text{eff}}(g)$ (top), strength $\widehat{\mathcal{C}}_g$ (middle), and TailPressure $\text{TP}_g$ (bottom) for representative high-leverage groups over training steps. (b) H-CES the same tracked quantities, with $\text{TP}_g$ constrained within the calibrated target band after warmup (green shading).

groups so that each group is probed at least once every 20k steps. This diagnostic adds less than 0.3% wall-clock time on the reference route and does not change optimization state because it uses `no_grad` and does not backpropagate.

### C.6. TailPressure Dynamics under Fixed $\pi_{\text{tr}}$

Even when $\pi_{\text{tr}}$ is time-homogeneous, TailPressure evolves because $\overline{w^2}_t(g)$ is stochastic and the EMA $e_{\text{eff}}(g)$ fluctuates, and because $\widehat{\mathcal{C}}_g$ is non-stationary under optimization. The diagnostic state is therefore time-varying and can be regulated in closed-loop. Figure 9 visualizes the two sources of non-stationarity and the effect of closed-loop regulation.

## D. Implementation and Matched-Budget Details

This section fixes the implementation choices behind the comparisons in Section 7. It defines Static-TP, explains what information is shared across methods, records grouping maps in Table 20, states the controller defaults evaluated in Table 6, and lists the matched-budget scan grids in Table 21.

### D.1. Static-TP Baseline

Static-TP sets a fixed per-group weight decay using contract-fixed leverage and pre-estimated exposure, without history buffers or controller ticks. Let $\bar{e}_g = \mathbb{E}_{\omega \sim \pi_{\text{tr}}}[w_g(\omega)^2]$. Static-TP sets

$$\lambda_g = \text{clip}\left(\lambda_0 \cdot \frac{\ell_{\text{tail}}(g)}{\bar{e}_g + \epsilon}, \lambda_{\min}, \lambda_{\max}\right), \tag{46}$$

with $(\lambda_0, \lambda_{\min}, \lambda_{\max})$ chosen by the same matched-budget scan and selection rule as H-CES.

### D.2. Fairness and Same-Information Usage

All methods can access $\omega = (b, q)$. Training-time usage includes environments for GroupDRO and diagnostic statistics for TailPressure. Evaluation-time usage includes conditional inference or calibration. The ablations Train-gated only and Eval-gated only in Table 4 isolate these effects under the same-information budget.

*Table 20.* Grouping summary for representative backbones used in the main tables. $\#g$ is the number of groups. %params is trainable parameter mass relative to the backbone. $\mathrm{med}(|\mathrm{Req}(g)|)$ is the median dependency set size. The full mapping table with layer ids is included in the released repository.

| Backbone / interface | $\#g$ | %params | $\mathrm{med}(|\mathrm{Req}(g)|)$ |
|---|---|---|---|
| LLaVA-NeXT/Vicuna-7B / LoRA | 112 | 0.20% | 2 |
| LLaVA-NeXT/Vicuna-7B / Adapters | 56 | 0.34% | 2 |
| LLaVA-NeXT/Vicuna-7B / Prefix | 28 | 0.16% | 2 |
| Qwen3-VL-8B / LoRA | 128 | 0.18% | 2 |
| MulT (HME) / fusion blocks | 32 | 0.42% | 1 |

*Table 21.* Matched-budget scan grids (representative; 36 configurations per method per route, 1 seed). All configurations train for the same number of optimizer steps with identical data pipelines and $\pi_{\mathrm{tr}}$.

| Method | Scanned hyperparameters (grid) |
|---|---|
| LoRA (global WD) | weight decay $\lambda \in \{0, 0.01, 0.03, 0.05, 0.08, 0.10\}$; LoRA rank $r \in \{8, 16\}$; LR multiplier $\eta \in \{0.5, 1.0, 2.0\}$ |
| LoRA + GroupDRO | same as LoRA plus GroupDRO $\alpha \in \{0.1, 0.2\}$; group smoothing $\epsilon \in \{0.01, 0.05\}$ |
| Static-TP | $\lambda_0 \in \{0.02, 0.04, 0.06, 0.08\}$; $\lambda_{\max} \in \{0.10, 0.15\}$; $\epsilon \in \{10^{-6}, 10^{-5}\}$; $r \in \{8, 16\}$ |
| H-CES | $\lambda_{\mathrm{init}} \in \{0.02, 0.04, 0.06\}$; $\lambda_{\max} \in \{0.10, 0.15\}$; $K \in \{100, 200\}$; $\gamma \in \{0.5, 0.8\}$; $r \in \{8, 16\}$ |

### D.3. Grouping Maps and Implementation Details

#### D.3.1. GROUPING SUMMARY

Injection points and mapping to groups $g \in \mathcal{G}$ are fixed per backbone. We report group counts, parameter mass per group, and dependency set sizes.

#### D.3.2. DEFAULT CONTROLLER HYPERPARAMETERS

Unless stated, $\rho = 0.05$, $K = 200$ steps, $\gamma = 0.5$, $\epsilon = 10^{-6}$, $e_{\mathrm{eff,min}} = 0.02$, $t_0 = 2000$ steps, and buffer length $L = 50$. Clipping bounds are $\lambda_{\min} = 0$ and route-dependent $\lambda_{\max}$. Table 6 in the main text varies these quantities one at a time around the default.

### D.4. Matched-Budget Scan Configurations

This appendix provides the scan ranges and matched-budget accounting used in Section 7.1. All methods use the same training steps and the same number of scan points per route.

## E. Extended Results and Stress Tests

This section collects appendix-only checks that complement Section 7. The full route table is reported in Table 2, and fixed-configuration seed robustness is reported in Table 3. The remaining appendix tables cover multi-seed counterfactuals, second-route counterfactual replication, marginal-gate checks, the non-PEFT example, stress-only metrics, boundary signatures, strength-proxy validation, overhead rules, and the selection rule.

### E.1. Extended Experiments

The experiments below are not used to replace the main headline protocol. They document whether the mechanism survives stricter counterfactuals, a second route, and a non-PEFT interaction surface.

#### E.1.1. FIXED-CONFIGURATION COUNTERFACTUAL SEED ROBUSTNESS

Table 22 repeats the core alignment counterfactuals from Table 4 over five fixed-configuration seeds. This appendix-only table addresses the seed-robustness concern without replacing the original single-run main-table protocol.

*Table 22.* Five-seed fixed-configuration counterfactuals on MMBench-EN dev v1.1 with LLaVA-NeXT/Vicuna-7B. Values are mean ± standard deviation.

| Method | Full | Worst-Head | Shift-Reweight |
|---|---|---|---|
| LoRA (tuned global WD) | 66.9 ± 0.2 | 45.8 ± 0.6 | 50.7 ± 0.5 |
| Static-TP (no loop) | 66.9 ± 0.2 | 50.1 ± 0.7 | 54.0 ± 0.6 |
| **H-CES (full)** | **67.0 ± 0.1** | **52.6 ± 0.6** | **56.2 ± 0.4** |
| Train-gated only | 66.9 ± 0.2 | 49.3 ± 0.6 | 53.3 ± 0.5 |
| Eval-gated only | 67.0 ± 0.2 | 46.9 ± 0.6 | 51.6 ± 0.4 |
| Shuffled $w$ (train-time) | 67.0 ± 0.2 | 46.5 ± 0.5 | 51.2 ± 0.4 |
| Permuted controller mapping | 67.0 ± 0.1 | 47.1 ± 0.6 | 51.8 ± 0.5 |
| Matched WD budget | 67.0 ± 0.2 | 47.9 ± 0.5 | 52.4 ± 0.4 |

*Table 23.* Counterfactual replication on a second route (MMMU val, LLaVA-NeXT/Vicuna-7B; score in %; single-run). All rows share the same contract objects and selection rule as Table 2.

| Method | Full | Worst-Head | Shift-Reweight |
|---|---|---|---|
| LoRA (global WD) | 35.8 | 21.4 | 26.5 |
| Static-TP (no loop) | 35.5 | 24.9 | 29.6 |
| H-CES (full) | 35.6 | **27.2** | **31.0** |
| shuffle-$w$ (train-time) | 35.6 | 22.0 | 27.0 |
| Permuted controller mapping | 35.6 | 22.5 | 27.6 |

### E.1.2. COUNTERFACTUAL REPLICATION

Table 23 replicates diagnostic counterfactuals on a second route under the same contract objects and selection rule.

### E.1.3. MARGINAL GATE MATCH METRICS

For a counterfactual gate $\tilde{w}$, we measure marginal match by

$$d_{\mathrm{marg}}(g) = \sum_{p \in \{0.1, 0.5, 0.9\}} |Q_p(w_g) - Q_p(\tilde{w}_g)|, \tag{47}$$

where $Q_p$ denotes the $p$-quantile over training batches. On the reference route, random-gate and shuffle-$w$ achieve median $d_{\mathrm{marg}}(g) \le 0.03$ across groups.

### E.2. Non-PEFT Conditional Interaction Example

This appendix instantiates the same diagnosis and control logic on a non-PEFT groupable interaction. We use a sparsely gated MoE fusion block inserted at the modality-fusion stage, with frozen expert weights and trainable router increments. Groups correspond to router increments per expert, and the gate is the observable health signal composed with the router top-$k$ selection. Exposure is measured by router selection frequencies aligned with Equation 15. Table 24 reports the resulting MOSI scores under the same closed-loop policy.

*Table 24.* Non-PEFT groupable interaction (MoE fusion block, MOSI; score in %; single-run). H-CES controls TailPressure for router increments using the same closed-loop policy.

| Method | Full | Train-Avg | Worst-Head | Shift-Reweight |
|---|---|---|---|---|
| MoE-fusion ERM | 84.8 | 79.0 | 66.2 | 70.5 |
| MoE-fusion + Static-TP | 84.6 | 79.1 | 72.3 | 75.2 |
| MoE-fusion + H-CES | 84.7 | 79.2 | **75.0** | **77.8** |

*Table 25.* Stress-only boundary tests on $\Omega_{\text{stress}}$. Reference route: MMBench-EN dev v1.1 with LLaVA-NeXT/Vicuna-7B. We report hard-missing score and output validity to separate accuracy loss from failure-to-respond.

| Method | Hard-missing score | Output validity | Vanishing-exposure Worst-Head |
|---|---|---|---|
| LoRA (global WD) | 33.4 | 0.68 | 44.8 |
| LoRA + GroupDRO | 35.1 | 0.71 | 45.5 |
| H-CES (full) | 38.0 | 0.78 | 46.0 |

### E.3. Training Overhead and Audit Footprint

This section documents the measurement protocol used to report the overhead and audit footprint of H-CES relative to the underlying increment interface. Table 7 reports representative overhead metrics on a reference route, and Appendix E.7 specifies the fixed hardware, warmup exclusion, averaging window, and log-volume accounting rules used to make these measurements comparable across methods.

### E.4. Stress-Only Metrics

Stress-only environments are $\Omega_{\text{stress}}$ and are excluded from headline claims. The definitions in Appendix E.4.1 specify what is measured on this boundary set, and Table 25 reports the corresponding stress-only results.

#### E.4.1. STRESS AGGREGATIONS

Let $\Omega_{\text{hard-miss}} \subseteq \Omega_{\text{stress}}$ denote environments with $b_m = 0$ for at least one modality and $q = 0$ for that modality. We define the hard-missing stress score as

$$\text{Score}_{\text{hard-miss}}(\theta) = \mathbb{E}_{\omega \sim \text{Unif}(\Omega_{\text{hard-miss}})}\big[\text{Score}_\omega(\theta)\big]. \tag{48}$$

We define an output-validity rate as

$$\text{ValidRate}_{\text{hard-miss}}(\theta) = \mathbb{E}_{\omega \sim \text{Unif}(\Omega_{\text{hard-miss}})}\Big[\mathbb{E}_{(x,y) \sim \mathcal{D}_\omega}\big[\mathbf{1}_{\{\text{ValidOut}(f_\theta(x,\omega))\}}\big]\Big], \tag{49}$$

where $\text{ValidOut}(\cdot)$ is a route-specific predicate.

#### E.4.2. STRESS-ONLY RESULTS

Table 25 is the numerical result for the stress-only metrics defined above. These values are not used as headline evidence for H-CES; they serve only to indicate boundary behavior when evaluation leaves the task-valid support used by Worst-Head and Shift-Reweight.

### E.5. Boundaries and Negative Results

#### E.5.1. VANISHING EXPOSURE YIELDS AN ESTIMATION TRADE-OFF

**Proposition 6** (Toy lower bound: conditional estimation scales as $1/(ne_{\text{eff}})$)**.** *Consider the scalar model $y = \theta^\star w + \epsilon$ where $\epsilon \sim \mathcal{N}(0,1)$ and $w \in \{0,1\}$ with $\Pr(w=1) = p$. Given $n$ i.i.d. samples, any unbiased estimator $\widehat{\theta}$ satisfies*

$$\mathbb{E}[(\widehat{\theta} - \theta^\star)^2] \geq \frac{1}{np}. \tag{50}$$

*For binary gates, $p = \mathbb{E}[w^2]$.*

*Proof.* Only samples with $w = 1$ carry information about $\theta^\star$. Let $N_1 = \sum_{i=1}^n w_i$ so $\mathbb{E}[N_1] = np$. Conditioned on $N_1 = k > 0$, Fisher information is $k$ and the Cramér–Rao bound gives $\text{Var}(\widehat{\theta} \mid N_1 = k) \geq 1/k$. Hence,

$$\text{Var}(\widehat{\theta}) = \mathbb{E}[\text{Var}(\widehat{\theta} \mid N_1)] \geq \mathbb{E}\left[\frac{1}{N_1}\right]. \tag{51}$$

*Table 26.* Boundary interpretation of TailPressure under severe missingness on MMBench-EN (dev v1.1). TailPressure is interpreted as mechanism evidence only in severe but headline-valid regimes; outside this regime it serves as a warning signal.

| Regime | In-regime fraction | Clip-event rate | $\rho(\text{TP}, \text{Worst-Head})$ | $\rho(\text{TP}, \text{Shift-Reweight})$ |
|---|---|---|---|---|
| Headline-valid severe | 86% | 3.2% | $-0.46$ | $-0.41$ |
| Vanishing exposure | 39% | 18.7% | $-0.18$ | $-0.12$ |
| Stress-only hard missing | 26% | 24.9% | $-0.08$ | $-0.04$ |

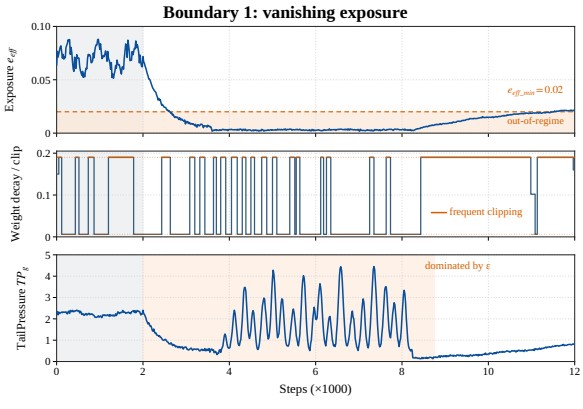

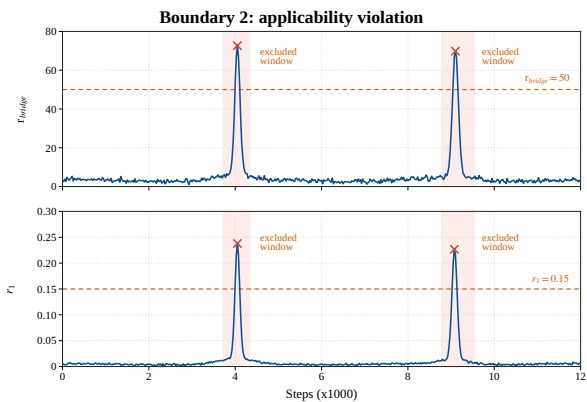

(a) Vanishing Exposure Causes Boundary Warnings

(b) Applicability Proxy Failures Mark Out-of-Regime Runs

*Figure 10.* Boundary signatures in logs. Vanishing exposure yields frequent actuator clipping and TailPressure dominated by $\epsilon$. Applicability failures appear as spikes in the proxy statistics beyond SSOT thresholds.

By Jensen inequality for the convex function $x \mapsto 1/x$ on $(0, \infty)$,

$$\mathbb{E}\left[\frac{1}{N_1}\right] \geq \frac{1}{\mathbb{E}[N_1]} = \frac{1}{np}. \tag{52}$$

$\square$

### E.5.2. MNAR NON-IDENTIFIABILITY

**Proposition 7** (MNAR: same observed distribution, different Full risk). *There exist two MNAR data-generating processes that induce the same observed distribution over $(x_{\text{obs}}, \omega, y)$ but different Full-environment risks $\mathcal{R}_{\omega_{\text{full}}}(\theta)$ for some predictors $\theta$. Therefore, without additional assumptions, no method relying only on observed data can guarantee Full risk under MNAR (Rubin, 1976; Little & Rubin, 2019).*

*Proof.* Construct a binary latent variable $u \in \{0, 1\}$ with label $y = u$. Let the observable input $x_{\text{obs}}$ be constant and let $b \in \{0, 1\}$ indicate whether the latent is observed. Define two MNAR mechanisms. Mechanism A sets $\Pr(b = 1 \mid y = 1) = 1$ and $\Pr(b = 1 \mid y = 0) = 0$ while mechanism B flips the dependence. With $\Pr(y = 1) = \Pr(y = 0) = 1/2$, both induce the same marginal frequency of $b$ and the same observed distribution over $(x_{\text{obs}}, b, y)$, but the conditional label distribution under the full condition differs, so $\mathcal{R}_{\omega_{\text{full}}}(\theta)$ differs for the same predictor $\theta$. $\square$

### E.5.3. RECOGNIZABLE BOUNDARY SIGNATURES

Table 26 and Figure 10 show the log signatures used to identify when TailPressure should be treated as a boundary warning rather than as mechanism evidence. This figure answers a diagnostic question: when TailPressure should not be interpreted as mechanism evidence, do log signatures make the failure mode recognizable without post-hoc labeling. Figure 10(a)–(b) highlight what to inspect in practice, namely actuator clipping events and proxy-threshold violations that define the audited regime.

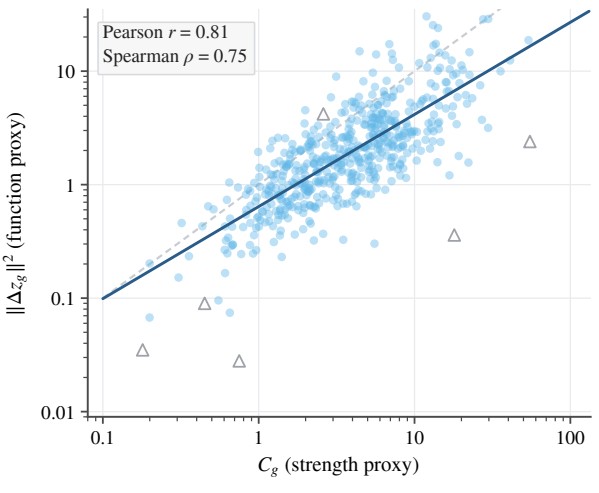

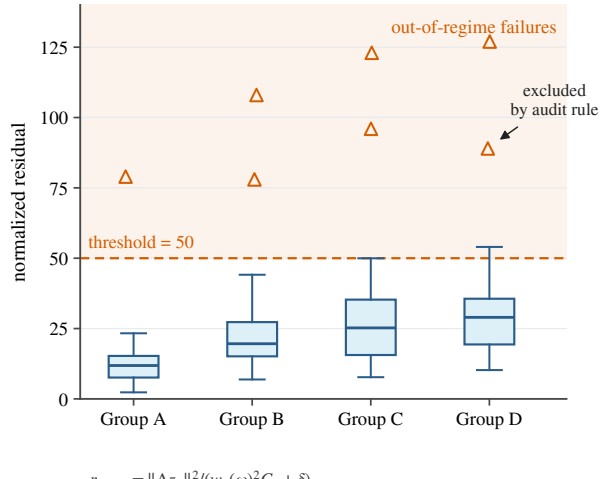

(a) Strength Proxy Correlates with Perturbation Size

(b) Strength Proxy Failure Modes Define Audit Boundaries

*Figure 11.* Strength proxy validation. We compare the parameter-norm proxy $\widehat{\mathcal{C}}_g$ to a function-level perturbation proxy computed by forward differencing of the loss input under controlled gating. The figure reports correlation and failure modes and motivates the audited regime thresholds.

### E.6. Strength Proxy Validation

Figure 11 validates the checkpoint-level strength proxy used in TailPressure against held-out perturbation measurements.

This section validates the use of the auditable parameter-norm proxy $\widehat{\mathcal{C}}_g$ as a strength statistic in the TailPressure state. Figure 11(a)–(b) indicate what to look for, namely correlation with a function-level perturbation proxy and identifiable failure modes that motivate the audited-regime thresholds.

### E.7. Overhead Measurement Rules

We measure overhead under fixed hardware and training settings. We fix batch size, sequence length, and gradient accumulation, exclude the first 100 warmup steps for timing, and report averages over 2000 consecutive post-warmup steps. Log volume is the compressed size of the audit map fields per 1000 steps.

### E.8. Selection Rule

We fix the selection rule used in all headline tables and controls. Let $\Delta_{\text{full}}$ be the allowed Full-drop tolerance relative to the baseline. Among configurations in the matched-budget scan that satisfy

$$\text{Score}_{\text{full}}(\theta) \geq \text{Score}_{\text{full}}(\theta_{\text{base}}) - \Delta_{\text{full}}, \tag{53}$$

we select the one with the largest $\text{Score}_{\text{tr-avg}}$. This rule is applied independently per route and method.

## F. Proofs and Additional Derivations

### F.1. Multi-Group Extension

Let $G$ denote the set of groups. For any distribution $\pi$ supported on $\Omega_{\text{head}}$, define the incremental sensitivity proxy

$$S_\pi(g) = \mathbb{E}_{\omega \sim \pi}[w_g(\omega)^2]\widehat{\mathcal{C}}_g. \tag{54}$$

Proposition 5 bounds the risk increment of toggling group $g$ by $K_g S_\pi(g)$. In multi-group models, interactions can invalidate naive additivity, but a falsifiable implication remains. If improvements are driven by a factor unrelated to gate alignment, then breaking alignment while preserving marginals should not remove both the tail gain and the TailPressure correlation. The counterfactuals in Table 4 and the bridge analysis in Appendix C.3 implement this test.

## F.2. Complete Proofs

### F.2.1. PROOF OF LEMMA 2

*Proof.* Let $z(\theta_g) = z_{\text{base}}(x, \omega) + w_g(\omega)\Phi_g(x, \omega)\theta_g$. By the chain rule,

$$\nabla_{\theta_g}\ell(z(\theta_g), y) = (\nabla_{\theta_g}z(\theta_g))^\top \nabla_z\ell(z(\theta_g), y). \tag{55}$$

Because $w_g$ and $\Phi_g$ do not depend on $\theta_g$,

$$\nabla_{\theta_g}z(\theta_g) = w_g(\omega)\Phi_g(x, \omega). \tag{56}$$

Substituting Equation 56 into Equation 55 yields Equation 18. Differentiating again and using that $w_g$ and $\Phi_g$ are constant with respect to $\theta_g$ gives Equation 19. $\square$

### F.2.2. SMOOTHNESS INCREMENT INEQUALITY

**Lemma 8** (Smoothness-based increment inequality). *Assume $\ell(\cdot, y)$ is $\beta$-smooth for all $y$ in the audited domain. Then for any prediction $z$ and perturbation $\Delta z$,*

$$\ell(z + \Delta z, y) \leq \ell(z, y) + \langle \nabla_z\ell(z, y), \Delta z \rangle + \frac{\beta}{2}\|\Delta z\|_2^2. \tag{57}$$

*Proof.* By $\beta$-smoothness, $\nabla_z^2\ell(\tilde{z}, y) \preceq \beta I$ along the segment between $z$ and $z + \Delta z$. Taylor theorem with integral remainder yields

$$\ell(z + \Delta z, y) = \ell(z, y) + \langle \nabla_z\ell(z, y), \Delta z \rangle + \int_0^1 (1 - t)\,\Delta z^\top \nabla_z^2\ell(z + t\Delta z, y)\,\Delta z\,dt, \tag{58}$$

and bounding the integrand by $\beta\|\Delta z\|_2^2$ gives the result. $\square$

### F.2.3. PROOF OF PROPOSITION 5

*Proof.* Fix group $g$ and distribution $\pi$ on $\Omega_{\text{head}}$. For any sample $(x, y, \omega)$, apply Lemma 8 with $z = z_{\text{base}}(x, \omega)$ and $\Delta z = \Delta z_g(x, \omega; \theta_g)$:

$$\ell(z_{\text{base}} + \Delta z_g, y) \leq \ell(z_{\text{base}}, y) + \langle \nabla_z\ell(z_{\text{base}}, y), \Delta z_g \rangle + \frac{\beta}{2}\|\Delta z_g\|_2^2. \tag{59}$$

Take expectations over $\omega \sim \pi$ and $(x, y) \sim \mathcal{D}_\omega$ and subtract the off risk:

$$\mathcal{R}_\pi(\text{with } g) - \mathcal{R}_\pi(\text{off } g) \leq \mathbb{E}\Big[\langle \nabla_z\ell(z_{\text{base}}, y), \Delta z_g \rangle\Big] + \frac{\beta}{2}\mathbb{E}[\|\Delta z_g\|_2^2]. \tag{60}$$

By Condition 4,

$$\left|\mathbb{E}\Big[\langle \nabla_z\ell(z_{\text{base}}, y), \Delta z_g \rangle\Big]\right| \leq \varepsilon_g^{(1)}\mathbb{E}[\|\Delta z_g\|_2^2]. \tag{61}$$

Hence,

$$\mathcal{R}_\pi(\text{with } g) - \mathcal{R}_\pi(\text{off } g) \leq \left(\varepsilon_g^{(1)} + \frac{\beta}{2}\right)\mathbb{E}[\|\Delta z_g\|_2^2]. \tag{62}$$

Apply Condition 3 pointwise:

$$\|\Delta z_g(x, \omega; \theta_g)\|_2^2 \leq w_g(\omega)^2 L_g^2 \widehat{\mathcal{C}}_g. \tag{63}$$

Taking expectations yields

$$\mathbb{E}[\|\Delta z_g\|_2^2] \leq L_g^2 \mathbb{E}[w_g(\omega)^2]\widehat{\mathcal{C}}_g. \tag{64}$$

Combining Equations 62 and 64 gives Equation 25. The specializations in Equations 26–28 follow by choosing $\pi = \pi_{\text{tr}}$, $\pi = \pi_{\text{sh}}$, and $\pi$ supported on $\Omega_{\text{worst}}$. $\square$

