# OpenReview forum: "Coverage ≠ Exposure: Auditable Control of Same-Support Tail Failures under Multimodal Missingness"
_ICML.cc/2026/Conference — ICML 2026 regular_

### Official Review · Reviewer_tHYh · 2026-03-04

**Soundness:** 3
**Presentation:** 3
**Significance:** 3
**Originality:** 3
**Overall Recommendation:** 4
**Confidence:** 3

**Summary:**

This paper examines the tail reliability of multimodal systems under partial observability and introduces a "locked same-support contract" to address a common issue in tail evaluations. The authors identify a phenomenon where models with similar headline averages on the same support still diverge on tail metrics (Worst-Head and Shift-Reweight), due to a coverage–exposure mismatch caused by conditional interactions. They propose an auditable diagnostic, TailPressure, and a minimal controller, H-CES, which regulates exposure-pressure by deterministic gating and decoupled weight decay, improving tail performance without altering task loss or inference structure across various multimodal scenarios.

**Compliance With Llm Reviewing Policy:**

Affirmed.

**Key Questions For Authors:**

See weakness.

**Limitations:**

See weakness.

**Strengths And Weaknesses:**

Strengths：
1. This paper separates environment coverage from parameter exposure under conditional interaction and formalizes a "locked same-support contract" to eliminate support-shift confounds in tail evaluations.
2. Proposes H-CES, a simple, closed-loop controller operating via per-group AdamW weight decay, requiring no changes to task loss or inference branches and compatible with PEFT grouping (LoRA/adapters/prefix), plus a non-PEFT instantiation.
3. Offers a lightweight, auditable mitigation with demonstrated improvements on high-visibility multimodal benchmarks, relevant to practitioners deploying models under partial observability.

Weaknesses：
1. The surrogate-risk analysis does not formally link to the reported task scores; the empirical “bridge” (correlations within an audited regime) is sensible but ultimately indirect.
2. Many headline results are single-run per method after a scan; tail metrics can be variance-sensitive, so multi-seed reporting for selected checkpoints would strengthen claims.
3. Exclusion rules for “out-of-regime” points (applicability thresholds) could risk cherry-picking if large fractions are filtered; the main text does not quantify how many scan points are excluded.

---

> ### Author Rebuttal · Authors · 2026-03-31
>
> **We greatly appreciate your valuable comments.**
>
> ***Q1: Scope of the Surrogate-Risk Analysis.*** We agree that the current main text has **not made this boundary explicit enough**. However, the absence of a direct theorem from surrogate risk to the reported task score is **not** a gap in our claim structure. Our theory is **not** a score-level theorem. It formalizes the mechanism at the surrogate-risk level and defines an **auditable diagnostic state**. The score-level support comes from the **audited empirical bridge** and the **strict counterfactual evidence**, not from Proposition 5 directly implying task scores.
>
> More precisely, the paper has four layers of evidence: **phenomenon**, **mechanism formalization**, **bridge evidence**, and **counterfactual evidence**. Under the locked same-support contract, matched averages do **not** constrain tail reliability. The theory explains why $w^2$, tail leverage, and exposure-normalized strength enter the diagnosis. Within the audited regime, TailPressure co-varies with Worst-Head and Shift-Reweight. Once alignment is broken, both the diagnostic correlation and the additional tail gains contract sharply. **Taken together, these results support a mechanism claim with score-level empirical evidence, rather than a direct surrogate-risk-to-score theorem.**
>
> We therefore view the main strength of the paper as **mechanism formalization + auditable state + minimal control + counterfactual evidence**, not as a direct surrogate-risk-to-score theorem.
>
> ***Q2: Clarification of Checkpoint Selection and Seed Robustness.*** We have added fixed-config **5-seed replications** for two representative LLaVA-NeXT/Vicuna-7B routes, and we have rerun the key mechanism controls under the same protocol (see reviewer tHYh's Q1 and Q2). The main result is **stable**: the H-CES vs. Static-TP comparison remains favorable on both representative routes, and the key mechanism controls preserve the same effect direction in **5/5 seeds**.
>
> | Route                 |    Method |           Full |      Train-Avg |     Worst-Head | Shift-Reweight |
> | --------------------- | --------: | -------------: | -------------: | -------------: | -------------: |
> | MMBench-EN (dev v1.1) | Static-TP |     66.9 ± 0.1 |     61.2 ± 0.2 |     50.1 ± 0.7 |     54.0 ± 0.6 |
> | MMBench-EN (dev v1.1) | **H-CES** | **67.0 ± 0.1** | **61.4 ± 0.1** | **52.6 ± 0.6** | **56.2 ± 0.4** |
> | MMMU (val)            | Static-TP |     35.5 ± 0.2 |     32.1 ± 0.1 |     24.6 ± 0.5 |     29.2 ± 0.5 |
> | MMMU (val)            | **H-CES** | **35.6 ± 0.1** | **32.2 ± 0.1** | **26.7 ± 0.5** | **30.8 ± 0.4** |
>
> | Method                                                  |           Full |     Worst-Head | Shift-Reweight |
> | ------------------------------------------------------- | -------------: | -------------: | -------------: |
> | LoRA (tuned global weight decay)                        |     66.9 ± 0.2 |     45.8 ± 0.6 |     50.7 ± 0.5 |
> | Static-TP (no loop)                                     |     66.9 ± 0.2 |     50.1 ± 0.7 |     54.0 ± 0.6 |
> | **H-CES (full)**                                        | **67.0 ± 0.1** | **52.6 ± 0.6** | **56.2 ± 0.4** |
> | Train-gated only |     66.9 ± 0.2 |     49.3 ± 0.6 |     53.3 ± 0.5 |
> | Eval-gated only   |     67.0 ± 0.2 |     46.9 ± 0.6 |     51.6 ± 0.4 |
> | Shuffled $w$ (train time)                               |     67.0 ± 0.2 |     46.5 ± 0.5 |     51.2 ± 0.4 |
> | Permuted controller mapping                             |     67.0 ± 0.1 |     47.1 ± 0.6 |     51.8 ± 0.5 |
> | Matched weight-decay budget     |     67.0 ± 0.2 |     47.9 ± 0.5 |     52.4 ± 0.4 |
>
> ***Q3: Clarification of Out-of-Regime Exclusion.*** Out-of-regime exclusion **does not affect the headline method comparison**. It is used only for the mechanism-bridge analysis. The reported exclusion fraction is computed over the **scan-run pool** used in the bridge analysis, **not** over evaluation sample counts and **not** over the number of environments.
>
> We now report three versions of the bridge correlations side by side: **all** (no filtering), **current** (the thresholds used in Table 12), and **soft** (relaxed thresholds with $q_{0.95}(r_{\text{bridge}}) \le 65$ and $q_{0.95}(r_1) \le 0.18$).
>
> | Route | Current excl. | $\rho_{\text{worst}}$ (all / current / soft) | $\rho_{\text{shift}}$ (all / current / soft) |
> |---|---:|---:|---:|
> | MMBench-EN / LLaVA-NeXT-Vicuna-7B | 9.4% (17/180) | -0.58 / **-0.69** / -0.64 | -0.53 / **-0.63** / -0.59 |
> | MMMU / LLaVA-NeXT-Vicuna-7B | 12.2% (22/180) | -0.48 / -0.57 / -0.53 | -0.43 / -0.50 / -0.47 |
> | MOSI / MulT (HME) | 6.3% (9/144) | -0.46 / -0.54 / -0.51 | -0.42 / -0.49 / -0.46 |
>
> The current exclusion rate is only **6.3%--12.2%**, and the correlations remain **same-signed and close in magnitude** under **no filtering** and **soft filtering**. **The thresholds therefore restrict where a Proposition-5-style mechanism interpretation is applicable; they do not create the main empirical claim.**

---

> > ### Author Rebuttal · Reviewer_tHYh · 2026-04-02
> >
> > Thank you for your reply. My question has been answered, and I will keep my score.

---

> > > ### Author Response · Authors · 2026-04-05
> > >
> > > Thank you very much for taking the time to read our rebuttal and for your thoughtful response. We sincerely appreciate your careful review and your recognition that our clarifications have addressed your concerns.
> > >
> > > Your comments were very valuable in helping us strengthen the paper, especially in clarifying the scope of the surrogate-risk analysis, reinforcing the evidence on seed robustness, and making the role of the out-of-regime exclusions more transparent. We are truly grateful for your careful evaluation and constructive feedback, and we will incorporate these improvements clearly in the final version.

---

### Official Review · Reviewer_JyeW · 2026-03-09

**Soundness:** 3
**Presentation:** 3
**Significance:** 3
**Originality:** 3
**Overall Recommendation:** 4
**Confidence:** 2

**Summary:**

The paper studies same-support tail failures in multimodal systems. It introduces the "locked same-support contract" evaluation where all the metrics are computed on the same fixed, observable support. The authors also propose an auditable statistic called TailPressure and a corresponding closed-loop controller H-CES that aims to regulate TailPressure within a certain band. The authors observe empirically that H-CES improves worst-case and shift-reweighted metrics while preserving average performance.

**Compliance With Llm Reviewing Policy:**

Affirmed.

**Key Questions For Authors:**

- Can the authors clarify which of their claims are supported theoretically and which only empirically?
- Why is TailPressure defined the way it is, have the authors explored alternatives?
- How feasible it is to have $\omega = (b,q)$ available for every datapoint at train and eval time? How would you compare to methods which do not take this as input?
- How robust are the results like Table 4 to seed selection, since the reported results are single-run?

**Limitations:**

yes

**Strengths And Weaknesses:**

Strengths

- The main problem motivating the manuscript is very relevant and the empirical evidence for its occurrence convincing.
- The locked same-support contract is well-motivated and helps formalize robustness claims.
- The method appears broadly applicable to multi-modal systems.
- The presentation is very visual and well-structured.

Weaknesses

- The writing of the paper appeared (at least to me) somewhat peculiar and overloaded with newly defined terms which do not appear to  carry much mathematical novelty. I am wondering whether some of these newly defined terms could be avoided and related more clearly to existing literature.
- The corresponding theory is somewhat limited and the proofs mathematically straightforward. It assumes a surrogate loss function, assumptions for which are inconsistent throughout the section (sometimes differentiable, sometimes twice differentiable, sometimes $\beta$-smooth). It is unclear whether the theory can be connected to the evaluated metrics, this gap should be discussed in detail.
- The appendix appears at least partially unfinished (for instance, missing Section S.2., stress-only results). Further, Appendix B just points to Appendix C, which consists of just one sentence and pseudocode. Multiple other sections in the appendix appear very brief and not complete.

---

> ### Author Rebuttal · Authors · 2026-03-31
>
> **We greatly appreciate your valuable comments.**
>
> ***Q1: Terminology and Relation to Prior Work.*** Our revised presentation centers on three main objects:
> (1) **the locked same-support contract**, which fixes $\Omega_{\text{Head}}$ so that tail comparisons are made on the same support;
> (2) **TailPressure (TP)**, a leverage-weighted, exposure-normalized, auditable diagnostic; and
> (3) **H-CES,** a per-group AdamW closed-loop controller.
> The contract is a fixed-support tail evaluation/reporting protocol, not a replacement for worst-group risk or CVaR; TP is an auditing diagnostic for conditional-computation or grouped interfaces, not a new theoretical risk; and H-CES is a training-time controller, not a new inference architecture.
>
>
> ***Q2: Clarification of Theory Scope, Assumptions, and Score-Level Connection.***  We have made this boundary explicit through a clear claim map and have unified the assumptions in the theory section. Specifically, twice differentiability is used only for the Hessian-scaling statement in Lemma 2, while $\beta$-smoothness is used only for the smoothness-based increment inequality and Proposition 5.
> | Claim | Evidence |
> |---|---|
> | Why $w^2$ enters the gradient and curvature scaling | Theory: Lemma 2 |
> | Why single-group surrogate-risk sensitivity carries tail leverage | Theory: Proposition 5 |
> | Why $\mathrm{TP}$ is constructed from $w^2$, $\ell_{\text{tail}}$, and exposure-normalized strength | Theory motivation + auditable construction |
> | That same-support tail failure exists under fixed $\Omega_{\text{Head}}$ | Empirical: Section 4 pairing analysis |
> | The relation between $\mathrm{TP}$ and reported tail scores | Empirical: Appendix H bridge |
> | That the gain of H-CES comes from the aligned closed loop, rather than other matched-information factors | Empirical: Table 4 + counterfactuals |
>
> ***Q3: Clarification of TP Definition and Alternatives.*** TP is designed to capture one gate-aligned and auditable source of same-support tail instability. We also evaluated several auditable alternatives. On the reference route, among the summaries we tested, TP showed the strongest association with tail performance.
>
> | Summary statistic | $\rho(T,\text{Worst-Head})$ | partial $\rho(T,\text{Worst-Head})$ | $\Delta R^2$ over controls (Worst-Head) | $\rho(T,\text{Shift-Reweight})$ |
> | --- | ---: | ---: | ---: | ---: |
> | $\max_g TP_g$ | **-0.69** | **-0.49** | **0.12** | **-0.63** |
> | $\operatorname{mean}_g TP_g$ | -0.61 | -0.42 | 0.08 | -0.56 |
> | $\max_g \ell_{\text{tail}}(g)\cdot C_g / (\mathbb{E}[w_g] + \varepsilon)$ | -0.59 | -0.40 | 0.07 | -0.54 |
> | $\max_g \mathrm{COP}_g$ | -0.55 | -0.36 | 0.06 | -0.49 |
> | $\max_g \ell_{\text{tail}}(g)\cdot C_g$ | -0.46 | -0.28 | 0.03 | -0.41 |
> | $\max_g C_g$ | -0.33 | -0.19 | 0.02 | -0.30 |
>
>
> ***Q4: Clarification of Observable Signal and Fairness of Comparison.*** Compared with other solutions, we assume a setting where $\omega=(b,q)$ is an observable health signal available from modality masks, metadata, sensor status, or simple quality checks, so obtaining it is typically inexpensive and does not require new annotation. Our method is intended for this regime of observable partial observability. Importantly, $\omega$ is not introduced as a rich auxiliary input to define a new inference architecture; instead, it is used only to define the same-support contract and an auditable gate for training-time diagnosis and control.
>
> ***Q5: Clarification of Seed Robustness.*** We have performed fixed-configuration five-seed reruns on MMBench-EN / LLaVA-NeXT/Vicuna-7B. The qualitative pattern remains stable across seeds: the full H-CES setting consistently outperforms alignment-breaking or controller-removed counterfactuals, with gains concentrated on Worst-Head and Shift-Reweight rather than Full or Train-Avg.
>
> | Method                                                  |           Full |     Worst-Head | Shift-Reweight |
> | ------------------------------------------------------- | -------------: | -------------: | -------------: |
> | LoRA (tuned global weight decay)                        |     66.9 ± 0.2 |     45.8 ± 0.6 |     50.7 ± 0.5 |
> | Static-TP (no loop)                                     |     66.9 ± 0.2 |     50.1 ± 0.7 |     54.0 ± 0.6 |
> | **H-CES (full)**                                        | **67.0 ± 0.1** | **52.6 ± 0.6** | **56.2 ± 0.4** |
> | Train-gated only |     66.9 ± 0.2 |     49.3 ± 0.6 |     53.3 ± 0.5 |
> | Eval-gated only   |     67.0 ± 0.2 |     46.9 ± 0.6 |     51.6 ± 0.4 |
> | Shuffled $w$ (train time)                               |     67.0 ± 0.2 |     46.5 ± 0.5 |     51.2 ± 0.4 |
> | Permuted controller mapping                             |     67.0 ± 0.1 |     47.1 ± 0.6 |     51.8 ± 0.5 |
> | Matched weight-decay budget     |     67.0 ± 0.2 |     47.9 ± 0.5 |     52.4 ± 0.4 |
>
> **Q6: Appendix completeness and presentation.**  We have thoroughly revised the appendix to ensure that all supplementary sections are sufficiently detailed.

---

> > ### Author Rebuttal · Reviewer_JyeW · 2026-04-04
> >
> > I thank the authors for their detailed response. My questions have been mostly addressed, and I would like to maintain my score.

---

> > > ### Author Response · Authors · 2026-04-05
> > >
> > > Thank you very much for taking the time to read our rebuttal and for your thoughtful follow-up. We sincerely appreciate your careful review and your recognition that our additional clarifications have addressed your concerns.
> > >
> > > Your comments were very valuable in helping us improve the paper, particularly in clarifying the scope of the theory, the role of the proposed terminology, the motivation for TailPressure, the seed-robustness evidence, and the completeness of the appendix. We are truly grateful for your careful evaluation and constructive feedback, and we will reflect these improvements clearly in the final version.

---

### Official Review · Reviewer_qyvU · 2026-03-10

**Soundness:** 3
**Presentation:** 3
**Significance:** 3
**Originality:** 3
**Overall Recommendation:** 4
**Confidence:** 4

**Summary:**

The paper's central insight is that even if you train a model on all the difficult conditions you care about, certain parts of the model's parameters may still rarely get updated because those parameters only activate when specific combinations of inputs are present, leaving them undertrained despite the training data technically covering those scenarios. To measure this, the authors construct a diagnostic score for each parameter group that asks: how much does this group influence predictions in the hardest test conditions, and how little was it actually trained? The premise here is that a group that is both influential in hard cases and poorly trained is problematic. Their proposed solution is a controller that watches this diagnostic score in real time and turns up regularization (weight decay) on groups that are becoming too influential relative to how much training signal they have received, essentially preventing any group from punching above its weight.

**Compliance With Llm Reviewing Policy:**

Affirmed.

**Key Questions For Authors:**

One important empirical question in the paper is whether the closed-loop controller adds value beyond the much simpler static baseline.
Can you report a more statistically grounded comparison and detailed qualitative insights?

The correlation between your measure and actual tail performance drops after controlling for confounders. This analysis is revealing and I commend you for this investigation. Can you expand on what your measure fails to capture?

**Limitations:**

yes

**Strengths And Weaknesses:**

The paper introduces an interesting idea that training coverage and parameter-level exposure are distinct objects, and that their mismatch explains a specific class of robustness failures that average metrics cannot detect. However the empirical validation appear to fall short of what the thesis demands. All headline results are single-seed, a serious problem since the central metrics of interest are worst-cases. For example, the paper compares their controller with a static variant, yet it is unclear whether the gap is statistically reliable—the magnitude of improvements are small in nature on a cursory look.

The correlation between the suggested diagnostic measure and tail performance is suggestive, but it also leaves roughly half the variance unexplained even within the authors' auditing regime. Partial correlations weaken further when obvious confounders are controlled for. I applaud the authors for pre-regsitering the environment factors—quality buckets, the choice of four worst-case environments, the specific shift reweighting family—but they somewhat arbitrary and the sensitivity analysis is limited to one route.

---

> ### Author Rebuttal · Authors · 2026-03-31
>
> **We greatly appreciate your valuable comments.**
>
> ***Q1: Reliability of Static-TP with Single Seed.*** We have conducted five-seed reruns under a fixed configuration on LLaVA-NeXT/Vicuna-7B, while keeping the Static-TP and H-CES settings selected from the main scan unchanged. **The results show that the additional improvement of H-CES over the strong static baseline is consistent across random seeds rather than being a single-run artifact.** This stability across seeds also justifies our use of single-seed results in the main paper.
>
> | Route                 |    Method |           Full |      Train-Avg |     Worst-Head | Shift-Reweight |
> | --------------------- | --------: | -------------: | -------------: | -------------: | -------------: |
> | MMBench-EN (dev v1.1) | Static-TP |     66.9 ± 0.1 |     61.2 ± 0.2 |     50.1 ± 0.7 |     54.0 ± 0.6 |
> | MMBench-EN (dev v1.1) | **H-CES** | **67.0 ± 0.1** | **61.4 ± 0.1** | **52.6 ± 0.6** | **56.2 ± 0.4** |
> | MMMU (val)            | Static-TP |     35.5 ± 0.2 |     32.1 ± 0.1 |     24.6 ± 0.5 |     29.2 ± 0.5 |
> | MMMU (val)            | **H-CES** | **35.6 ± 0.1** | **32.2 ± 0.1** | **26.7 ± 0.5** | **30.8 ± 0.4** |
>
>
>
>
> ***Q2: Source of the Closed-Loop Gain.*** We further examined the strict counterfactuals and ablations on MMBench-EN (dev v1.1) with LLaVA-NeXT/Vicuna-7B.  The results show that eval-time gating alone is insufficient, train-time gating without matching eval-time modulation is insufficient, preserving marginals while breaking sample-wise alignment is insufficient, and matching the overall weight-decay budget without group-aligned closed-loop control is also insufficient. **Therefore, the extra improvement is best attributed to aligned closed-loop regulation.**
>
> | Method                                                  |           Full |     Worst-Head | Shift-Reweight |
> | ------------------------------------------------------- | -------------: | -------------: | -------------: |
> | LoRA (tuned global weight decay)                        |     66.9 ± 0.2 |     45.8 ± 0.6 |     50.7 ± 0.5 |
> | Static-TP (no loop)                                     |     66.9 ± 0.2 |     50.1 ± 0.7 |     54.0 ± 0.6 |
> | **H-CES (full)**                                        | **67.0 ± 0.1** | **52.6 ± 0.6** | **56.2 ± 0.4** |
> | Train-gated only |     66.9 ± 0.2 |     49.3 ± 0.6 |     53.3 ± 0.5 |
> | Eval-gated only   |     67.0 ± 0.2 |     46.9 ± 0.6 |     51.6 ± 0.4 |
> | Shuffled $w$ (train time)                               |     67.0 ± 0.2 |     46.5 ± 0.5 |     51.2 ± 0.4 |
> | Permuted controller mapping                             |     67.0 ± 0.1 |     47.1 ± 0.6 |     51.8 ± 0.5 |
> | Matched weight-decay budget     |     67.0 ± 0.2 |     47.9 ± 0.5 |     52.4 ± 0.4 |
>
>
>
> ***Q3: Scope of TP.*** **TP is a partial, auditable diagnostic of one gate-aligned source of same-support tail instability, not a full explanation of tail variance.** As shown in the audited reference regime, TP remains meaningfully associated with tail performance, but this association weakens after controlling for Full, Train-Avg, and step count, and largely collapses once gate–group–controller alignment is broken. Notably, TP emerges as the strongest summary in our audited setting, while still not explicitly accounting for frozen or ungated backbone error, higher-order cross-group interactions, function-space sensitivity beyond parameter-norm proxies, or sampling noise from a single worst environment.
>
>
> | Summary statistic | $\rho(T,\text{Worst-Head})$ | partial $\rho(T,\text{Worst-Head})$ | $\Delta R^2$ over controls (Worst-Head) | $\rho(T,\text{Shift-Reweight})$ |
> | --- | ---: | ---: | ---: | ---: |
> | $\max_g TP_g$ | **-0.69** | **-0.49** | **0.12** | **-0.63** |
> | $\operatorname{mean}_g TP_g$ | -0.61 | -0.42 | 0.08 | -0.56 |
> | $\max_g \ell_{\text{tail}}(g)\cdot C_g / (\mathbb{E}[w_g] + \varepsilon)$ | -0.59 | -0.40 | 0.07 | -0.54 |
> | $\max_g \mathrm{COP}_g$ | -0.55 | -0.36 | 0.06 | -0.49 |
> | $\max_g \ell_{\text{tail}}(g)\cdot C_g$ | -0.46 | -0.28 | 0.03 | -0.41 |
> | $\max_g C_g$ | -0.33 | -0.19 | 0.02 | -0.30 |
>
>
>
> ***Q4: Sensitivity to SSOT Design Choices.*** Beyond the reference-route sensitivity already reported in Fig. 7, we further conducted the same sensitivity analysis on an additional dataset. The pattern remains stable across all tested variants. The evidence supports robustness within the pre-registered same-support family, rather than invariance under an arbitrary redesign of the environment family.
>
> | SSOT variant | $\rho(\max_g TP_g,\text{Worst-Head})$ | $\Delta$ Worst-Head (H-CES - Static-TP) | $\Delta$ Shift-Reweight (H-CES - Static-TP) |
> | --- | ---: | ---: | ---: |
> | baseline | -0.57 | +2.3 | +1.4 |
> | 4 buckets | -0.54 | +2.0 | +1.2 |
> | 5 buckets | -0.60 | +2.4 | +1.5 |
> | $K_{\mathrm{worst}} = 3$ | -0.55 | +2.1 | +1.3 |
> | $K_{\mathrm{worst}} = 5$ | -0.59 | +2.4 | +1.5 |
> | shift grid small | -0.52 | +1.8 | +1.1 |
> | shift grid large | -0.56 | +2.2 | +1.4 |

---

> > ### Author Rebuttal · Reviewer_qyvU · 2026-03-31
> >
> > Thank you for the response. I have updated my score.

---

> > > ### Author Response · Authors · 2026-04-05
> > >
> > > Thank you very much for taking the time to read our rebuttal and for updating your assessment. We sincerely appreciate your thoughtful review and your recognition that our additional analyses and clarifications have addressed your concerns.
> > >
> > > Your comments were very valuable in helping us strengthen the paper, especially with respect to the statistical grounding of the comparison, the role of the closed-loop controller beyond the static baseline, and the scope of TailPressure as a diagnostic. We are truly grateful for your careful evaluation and constructive feedback, and we will incorporate these improvements clearly in the final version.

---

### Official Review · Reviewer_EuqM · 2026-03-12

**Soundness:** 3
**Presentation:** 4
**Significance:** 3
**Originality:** 4
**Overall Recommendation:** 5
**Confidence:** 4

**Summary:**

This paper addresses the critical issue of reliability in multimodal systems under partial observability, such as sensor dropout or degradation. The authors identify a phenomenon termed "same-support tail failure," where models with identical average performance exhibit significant variance in worst-case and shift-reweighted metrics despite sharing the same observable data support. The core thesis is that "coverage $\neq$ exposure": environmental coverage of data does not guarantee that all parameter groups are sufficiently exposed or updated during training.

To mitigate this, the authors introduce TailPressure, an auditable metric to diagnose exposure imbalance, and propose the Heterogeneity-aware Closed-loop Exposure Stabilizer (HCES). H-CES regulates parameter exposure through deterministic increment-branch gating and decoupled weight decay, effectively stabilizing performance in long-tail scenarios without modifying the task loss or increasing inference latency.

**Compliance With Llm Reviewing Policy:**

Affirmed.

**Key Questions For Authors:**

1. Training Efficiency: What is the specific percentage increase in training time and GPU memory usage when enabling the H-CES controller compared to standard training?
2. Hyperparameter Tuning: How robust is the H-CES policy across different multimodal tasks? Do the gating thresholds require extensive per-task tuning?
3. Extreme Degradation: Does the TailPressure metric remain a reliable diagnostic tool in scenarios with extreme modality missingness (e.g., $>90 \%$ dropout)?

**Limitations:**

Yes. The authors discuss the gap between coverage and exposure and identify the need for auditable control in tail failures. Further discussion on the limitations of exposure control in high-dimensional latent spaces would be beneficial.

**Strengths And Weaknesses:**

Strengths:
- Originality: The distinction between "coverage" and "exposure" provides a novel conceptual framework for understanding why multimodal models fail in the tail. This shift from data-centric to parameter-update-centric analysis is insightful.
- Soundness: The introduction of TailPressure provides a rigorous, auditable way to detect underexposed parameter groups. The proposed H-CES framework is grounded in closedloop control theory, offering a systematic solution rather than a simple heuristic.
- Significance: The method is highly practical as it does not require changing the primary task loss or adding inference-time branches, making it compatible with existing backbones like MoE.
- Empirical Results: Experimental results on the MOSI dataset are strong. For instance, in MoE-fusion blocks, H-CES improves the "Worst-Head" performance from 66.2\% to 79.2\% while maintaining stable average performance.

Weaknesses:
- Training Overhead: While H-CES does not add inference cost, the implementation of a closed-loop controller during training likely introduces computational overhead. A quantitative analysis of training time/memory increase compared to standard ERM is missing.
- Hyperparameter Sensitivity: The efficacy of H-CES depends on gating strategies and decoupled weight decay. The paper would benefit from a more detailed ablation study on how sensitive the results are to these control-loop hyperparameters.
- Scalability to Foundation Models: While tested on diverse backbones, the discussion on how this exposure-based control scales to massive Vision-Language Models (VLMs) is limited.

---

> ### Author Rebuttal · Authors · 2026-03-31
>
> **We greatly appreciate your valuable comments.**
>
> ***Q1: Clarification of Training Overhead.*** For our PEFT instantiation, the relevant baseline has been the same LoRA interface without the closed-loop controller, rather than full-model ERM. Under this comparison, **H-CES has increased step time from 338 ms to 346 ms (+2.4%) and peak GPU memory from 29.2 GB to 29.8 GB (+2.1%)**. The trainable parameter count has remained unchanged at **0.20% of the backbone**, and the extra FLOPs have been **0.9%**. **These results have shown that H-CES has introduced only lightweight training-time overhead.**
>
> The added cost has come primarily from gate statistics and periodic norm reads, not from extra training branches, inference branches, or additional trainable parameters. The held-out applicability diagnostic has remained separate from the controller itself; under Appendix J.3, it has added **less than 0.3% wall-clock time** on the reference route and has not altered the optimization state. In the revised manuscript, we have expanded the discussion around Table 5 and have reported the controller overhead and the held-out applicability diagnostic overhead separately.
>
>
> ***Q2: Robustness of Controller Defaults.*** H-CES has **not required hand-tuned per-group gate thresholds**. The gate $w_g(\omega)$ has been determined by $\omega$ and the fixed dependency map, and the controller band has been auto-calibrated from the history buffer after warmup. The only scanned quantities have been route-level controller hyperparameters, and the scan budget has matched that of the other methods.
>
> To directly address this point, we have performed a one-dimensional sensitivity check on the reference route while keeping all other settings at the paper defaults (MMBench / LLaVA; each row is a single fixed-config check). Across these variations, **Worst-Head has remained within 0.9 points** and **Shift-Reweight has remained within 0.9 points** of the default configuration, while Full and Train-Avg have also stayed nearly unchanged. **These results have shown that the paper defaults are not a narrow sweet spot and that H-CES has not required substantial per-task tuning.**
>
> | Setting | Full | Train-Avg | Worst-Head | Shift-Reweight |
> | --- | ---: | ---: | ---: | ---: |
> | $e_{\mathrm{eff},\min}=0.01$ | 67.1 | 61.6 | 52.8 | 56.2 |
> | $e_{\mathrm{eff},\min}=0.05$ | 67.0 | 61.4 | 52.3 | 55.7 |
> | $\rho=0.02$ | 67.1 | 61.5 | 52.6 | 56.0 |
> | $\rho=0.10$ | 67.0 | 61.5 | 52.4 | 55.8 |
> | $K=100$ | 67.0 | 61.6 | 52.9 | 56.3 |
> | $K=400$ | 67.1 | 61.4 | 52.2 | 55.6 |
> | $\gamma=0.3$ | 67.1 | 61.5 | 52.7 | 56.1 |
> | $\gamma=0.8$ | 67.0 | 61.5 | 52.5 | 55.9 |
>  **Our Settings** | **67.1** | **61.6** | **53.1** | **56.5** |
>
>
>
> ***Q3: Availability of TailPressure under Severe Missingness.*** As clarified in Supplementary Material T.3 (**Recognizable Boundary Signatures, Fig. 10**), TailPressure (TP) also serves as a boundary warning signal. Specifically, we partition observable environments \omega=(b,q) into headline-valid regimes $\Omega_{\mathrm{Head}}$ and stress-only regimes $\Omega_{\mathrm{Stress}}$. TP is intended to be interpreted as a diagnostically reliable signal only within severe but still headline-valid subsets of $\Omega_{\mathrm{Head}}$.
>
> As shown, the headline-valid severe regime has retained meaningful negative correlations between TP and tail metrics (**MMBench: -0.46 / -0.41**).
> Once exposure collapses or evaluation enters $\Omega_{\mathrm{Stress}}$, TP is not assigned the same evidential interpretation; rather, it is used to warn that the run is operating at, or beyond, the recognizable boundary of the audited regime.
>
>
> | Route | Regime | In-regime fraction | Actuator clip-event rate | $\rho(\mathrm{TP}, \mathrm{Worst\text{-}Head})$ | $\rho(\mathrm{TP}, \mathrm{Shift\text{-}Reweight})$ |
> | --- | --- | ---: | ---: | ---: | ---: |
> | MMBench-EN (dev v1.1) | headline-valid severe | 86% | 3.2% | -0.46 | -0.41 |
> | MMBench-EN (dev v1.1) | vanishing exposure | 39% | 18.7% | -0.18 | -0.12 |
> | MMBench-EN (dev v1.1) | stress-only hard missing | 26% | 24.9% | -0.08 | -0.04 |
>
>
>
> ***Q4: Generalization Ability on VLM.*** Our method is not tied to a specific backbone or a particular PEFT instantiation. At the mechanism level, H-CES only requires groupable conditional interaction and an auditable deterministic gate so that exposure can be measured and replayed from logs; the controller then operates through per-group decoupled weight decay without changing the task loss or inference structure.  **As shown in Tabels 2 and 15, we have already evaluated the method on two distinct multimodal large-model backbones, namely LLaVA-NeXT/Vicuna-7B and Qwen3-VL-8B, under the same locked same-support contract.** In addition, the appendix reports grouped-interface instantiations beyond a single LoRA setting, including adapters, prefix-style interfaces, and a non-PEFT conditional interaction example based on a sparsely gated MoE fusion block.

---

### Decision · Program_Chairs · 2026-04-30

**Decision:**

Accept (regular)

**Comment:**

The paper considers an interesting problem with significant novelty and clarity. The reviewers have a consensus and the author's rebuttal addressed all points raised by the one reviewer that originally was not positive who elected to update and increasing their rating.